# Using the past to estimate sensory uncertainty

Ulrik Beierholm[1†]*, Tim Rohe[2,3†], Ambra Ferrari[4], Oliver Stegle[5,6,7], Uta Noppeney[4,8]

[1]Psychology Department, Durham University, Durham, United Kingdom; [2]Department of Psychiatry and Psychotherapy, University of Tübingen, Tübingen, Germany; [3]Department of Psychology, Friedrich-Alexander University Erlangen-Nuernberg, Erlangen, Germany; [4]Centre for Computational Neuroscience and Cognitive Robotics, University of Birmingham, Birmingham, United Kingdom; [5]Max Planck Institute for Intelligent Systems, Tübingen, Germany; [6]European Molecular Biology Laboratory, Genome Biology Unit, Heidelberg, Germany; [7]Division of Computational Genomics and Systems Genetics, German Cancer Research Center (DKFZ), Heidelberg, Germany, Heidelberg, Germany; [8]Donders Institute for Brain, Cognition and Behaviour, Radboud University, Nijmegen, Netherlands

*For correspondence:
ulrik.beierholm@durham.ac.uk

[†]These authors contributed equally to this work

Competing interests: The authors declare that no competing interests exist.

**Abstract** To form a more reliable percept of the environment, the brain needs to estimate its own sensory uncertainty. Current theories of perceptual inference assume that the brain computes sensory uncertainty instantaneously and independently for each stimulus. We evaluated this assumption in four psychophysical experiments, in which human observers localized auditory signals that were presented synchronously with spatially disparate visual signals. Critically, the visual noise changed dynamically over time continuously or with intermittent jumps. Our results show that observers integrate audiovisual inputs weighted by sensory uncertainty estimates that combine information from past and current signals consistent with an optimal Bayesian learner that can be approximated by exponential discounting. Our results challenge leading models of perceptual inference where sensory uncertainty estimates depend only on the current stimulus. They demonstrate that the brain capitalizes on the temporal dynamics of the external world and estimates sensory uncertainty by combining past experiences with new incoming sensory signals.

## Introduction

Perception has been described as a process of statistical inference based on noisy sensory inputs (*Knill and Pouget, 2004*; *Knill and Richards, 1996*). Key to this perceptual inference is the estimation and/or representation of sensory uncertainty (as measured by variance, i.e. the inverse of reliability/precision). Most prominently, in multisensory perception, a more reliable or 'Bayes-optimal' percept is obtained by integrating sensory signals that come from a common source weighted by their relative reliabilities with less weight assigned to less reliable signals. Likewise, sensory uncertainty shapes observers' causal inference. It influences whether observers infer that signals come from a common cause and should hence be integrated or else be processed independently (*Aller and Noppeney, 2019*; *Körding et al., 2007*; *Rohe et al., 2019*; *Rohe and Noppeney, 2015b*; *Rohe and Noppeney, 2015a*; *Rohe and Noppeney, 2016*; *Wozny et al., 2010*; *Acerbi et al., 2018*). Indeed, accumulating evidence suggests that human observers are close to optimal in many perceptual tasks (though see *Acerbi et al., 2014*; *Drugowitsch et al., 2016*; *Shen and Ma, 2016*; *Meijer et al., 2019*) and weight signals approximately according to their

 

sensory reliabilities (*Alais and Burr, 2004*; *Ernst and Banks, 2002*; *Jacobs, 1999*; *Knill and Pouget, 2004*; *van Beers et al., 1999*; *Drugowitsch et al., 2014*; *Hou et al., 2019*).

An unresolved question is how human observers compute their sensory uncertainty. Current theories and experimental approaches generally assume that observers access sensory uncertainty near-instantaneously and independently across briefly ($\leq$200 ms) presented stimuli (*Ma and Jazayeri, 2014*; *Zemel et al., 1998*). At the neural level, theories of probabilistic population coding have suggested that sensory uncertainty may be represented instantaneously in the gain of the neuronal population response (*Ma et al., 2006*; *Hou et al., 2019*). Yet, in our natural environment, sensory noise often evolves at slow timescales. For instance, visual noise slowly varies when walking through a snow storm. Observers may capitalize on the temporal dynamics of the external world and use the past to inform current estimates of sensory uncertainty. In this alternative account, more reliable estimates of sensory uncertainty would be obtained by combining past estimates with current sensory inputs as predicted by Bayesian learning.

To arbitrate between these two critical hypotheses, we presented observers with audiovisual signals in synchrony but with a small spatial disparity in a sound localization task. Critically, the spatial standard deviation (STD) of the visual signal changed dynamically over time continuously (experiments 1–3) or discontinuously (i.e. with intermittent jumps; experiment 4). First, we investigated whether the influence of the visual signal location on observers' perceived sound location depended on the noise only of the current visual signal or also of past visual signals. Second, using computational modeling and Bayesian model comparison, we formally assessed whether observers update their visual uncertainty estimates consistent with (i) an instantaneous learner, (ii) an optimal Bayesian learner, or (iii) an exponential learner.

## Results

In a spatial localization task, we presented participants with audiovisual signals in a series of four experiments, in which the physical visual noise changed dynamically over time either continuously or discontinuously (*Figure 1*). Visual (V) signals (clouds of 20 bright dots) were presented every 200 ms for a duration of 32 ms. The cloud's horizontal STD varied over time at this temporal rate of 5 Hz either continuously (experiments 1–3) or discontinuously with intermittent jumps (experiment 4). The cloud's location mean was temporally independently resampled from five possible locations ($-10°$, $-5°$, $0°$, $5°$, $10°$) on each trial with the inter-trial asynchrony jittered between 1.4 and 2.8 s. In synchrony with the change in the cloud's mean location, the dots changed their color and a sound was presented (AV signal). The location of the sound was sampled from the two possible locations adjacent to the visual cloud's mean location (i.e. $\pm5°$ AV spatial disparity). Participants localized the sound and indicated their response using five response buttons.

The small audiovisual disparity enabled an influence of the visual signal location on the perceived sound location as a function of visual noise (*Alais and Burr, 2004*; *Battaglia et al., 2003*; *Meijer et al., 2019*). As a result, observers' visual uncertainty estimate could be quantified in terms of the relative weight of the auditory signal on the perceived sound location with a greater auditory weight indicating that observers estimated a greater visual uncertainty.

In the first three experiments, we used continuous sequences, where the visual cloud's STD changed periodically according to a sinusoid (n = 25; period = 30 s), a random walk (RW1; n = 33; period = 120 s) or a smoothed random walk (RW2; n = 19; period = 30 s; *Figure 2*). In an additional fourth experiment, we inserted abrupt increases or decreases into a sinusoidal evolution of the visual cloud's STD (n = 18, period = 30 s, Figure 5). We will first describe the results for the three continuous sequences followed by the discontinuous sequence.

We assigned the sound localization responses and the associated physical visual noise (i.e. the cloud's STD) to 20 (resp. 15 for experiment 4) temporally adjacent bins covering the entire period of each of the three sequences. Each experiment repeated the same 30 s (Sin, RW2) or 120 s (RW1) period throughout the experiment resulting in ~32 periods for the RW1 and ~130 periods for the Sin and RW2 sequences. The trial and hence sound onsets were jittered with respect to this periodic evolution of the visual cloud's STD resulting in a greater effective sampling rate than expected for an inter-trial asynchrony of 1.4–2.8 s. In total, we assigned at least 44–87 trials to each bin (*Supplementary file 1-Table 1*). We quantified the auditory and visual influence on observers'

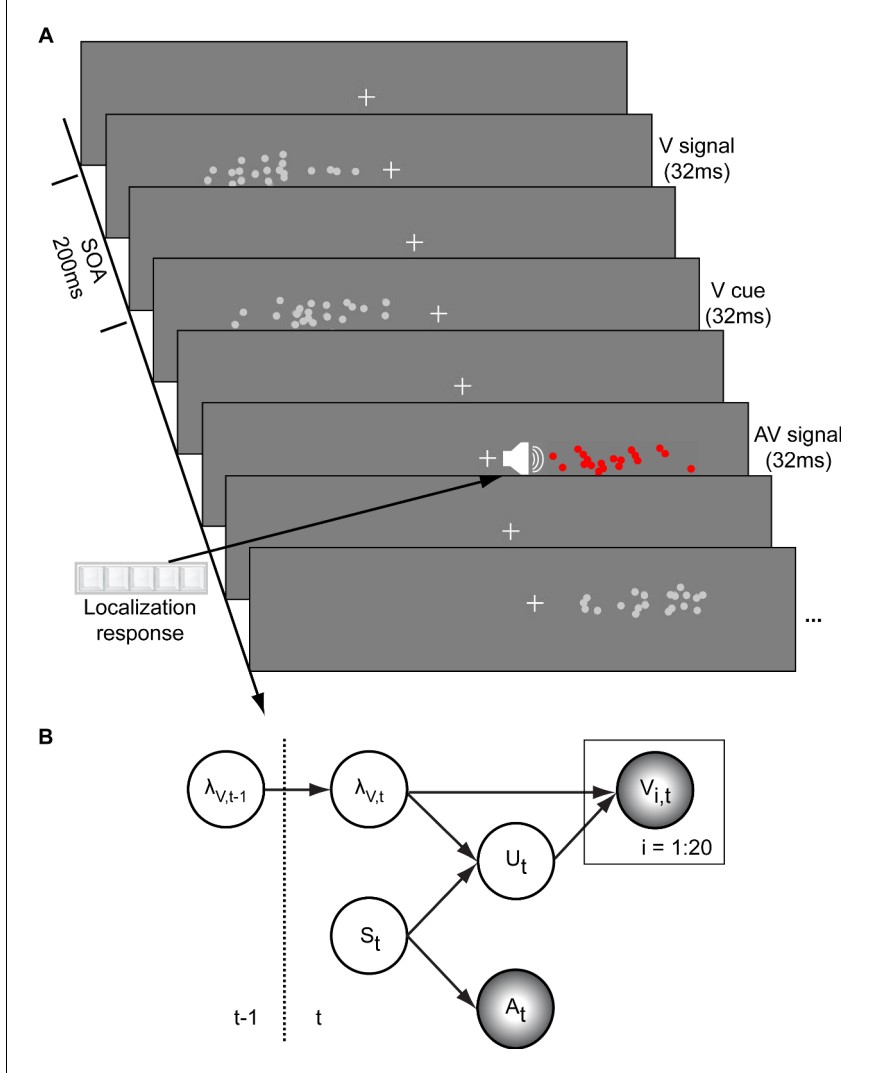

**Figure 1.** Audiovisual localization paradigm and Bayesian causal inference model for learning visual reliability. (**A**) Visual (V) signals (cloud of 20 bright dots) were presented every 200 ms for 32 ms. The cloud's location mean was temporally independently resampled from five possible locations ($-10°$, $-5°$, $0°$, $5°$, $10°$) with an inter-trial asynchrony jittered between 1.4 and 2.8 s. In synchrony with the change in the cloud's mean location, the dots changed their color and a sound was presented (AV signal) which the participants localized using five response buttons. The location of the sound was sampled from the two possible locations adjacent to the visual cloud's mean location (i.e. $\pm5°$ AV spatial). (**B**) The generative model for the Bayesian learner explicitly modeled the potential causal structures, that is whether visual ($V_i$) signals and an auditory (A) signal were generated by one common audiovisual source $S_t$, that is C = 1, or by two independent sources $S_{Vt}$ and $S_{At}$, that is C = 2 (n.b. only the model component for the common source case is shown to illustrate the temporal updating, for complete generative model, see *Figure 1—figure supplement 1*). Importantly, the reliability (i.e. 1/variance) of the visual signal at time t ($\lambda_t$) depends on the reliability of the previous visual signal ($\lambda_{t-1}$) for both model components (i.e. common and independent sources).

The online version of this article includes the following figure supplement(s) for figure 1:

**Figure supplement 1.** Generative model for the Bayesian learner.

---

perceived auditory location for each bin based on regression models (separately for each of the 20 temporally adjacent bins). For instance, for bin = 1 we computed:

$$R_{A,trial,bin=1} = L_{A,trial,bin=1}\beta_{A,bin=1} + L_{V,trial,bin=1}\beta_{V,bin=1} + \beta_{const,bin=1} + e_{trial,bin=1}$$

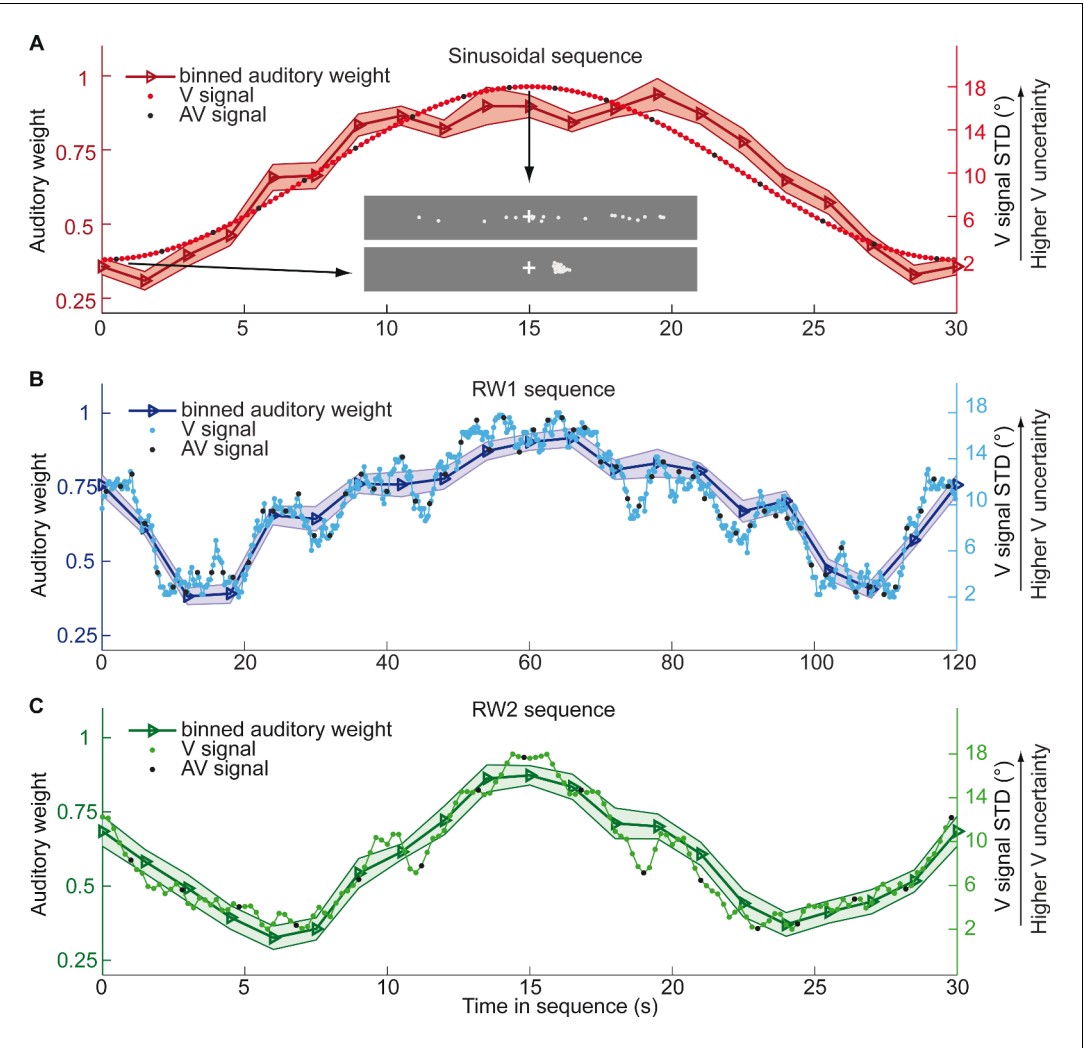

**Figure 2.** Time course of visual noise and relative auditory weights for continuous sequences of visual noise. The visual noise (i.e. STD of the cloud of dots, right ordinate) and the relative auditory weights (mean across participants ± SEM, left ordinate) are displayed as a function of time. The STD of the visual cloud was manipulated as (**A**) a sinusoidal (period 30 s, N = 25), (**B**) a random walk (RW1, period 120 s, N = 33) and (**C**) a smoothed random walk (RW2, period 30 s, N = 19). The overall dynamics as quantified by the power spectrum is faster for RW2 than RW1 (peak in frequency range [0 0.2] Hz: Sinusoid: 0.033 Hz, RW1: 0.025 Hz, RW2: 0.066 Hz). The RW1 and RW2 sequences were mirror-symmetric around the half-time (i.e. the second half was the reversed first half). The visual clouds were re-displayed every 200 ms (i.e. at 5 Hz). The trial onsets, that is audiovisual (AV) signals (color change with sound presentation, black dots), were interspersed with an inter-trial asynchrony jittered between 1.4 and 2.8 s. On each trial observers located the sound. The relative auditory weights were computed based on regression models for the sound localization responses separately for each of the 20 temporally adjacent bins that cover the entire period within each participant. The relative auditory weights vary between one (i.e. pure auditory influence on the localization responses) and zero (i.e. pure visual influence). For illustration purposes, the cloud of dots for the lowest (i.e. V signal STD = 2°) and the highest (i.e. V signal STD = 18°) visual variance are shown in (**A**).

The online version of this article includes the following figure supplement(s) for figure 2:

**Figure supplement 1.** Time course of the relative auditory weights for continuous sequences of visual noise when controlling for location of the cloud of dots in the previous trial.

with $R_{A,trial,bin=1}$ = Localization response for trial t and bin 1; $L_{A,trial,bin=1}$ or $L_{V,trial,bin=1}$ = 'true' auditory or visual location for trial t and bin 1; $\beta_{A,bin=1}$ or $\beta_{V,bin=1}$ = auditory or visual weight for bin 1; $\beta_{const,bin=1}$ = constant term; $e_{trial,bin=1}$ = error term. For each bin b, we thus obtained one auditory and one visual weight estimate. The relative auditory weight for a particular bin was computed as $w_{A,bin}$ = $\beta_{A,bin}$ / ($\beta_{A,bin}$ + $\beta_{V,bin}$).

*Figure 2* and *Figure 3* show the temporal evolution of the STD of the physical visual noise and observers' relative auditory weight indices $w_{A,bin}$. If observers estimate sensory uncertainty instantaneously, observer's relative auditory weight indices should closely track the visual cloud's STD (*Figure 2*). By contrast, we observed systematic biases: while the temporal evolution of the physical visual noise was designed to be symmetrical for each time period, we observed a temporal asymmetry for $w_A$ in all of the three experiments. For the monotonic sinusoidal sequence, $w_A$ was smaller for the 1st half of each period, when visual noise increased, than the 2nd half, when visual noise decreased over time (*Figure 3A*). For the non-monotonic RW1 and RW2 sequences, we observed more complex temporal profiles, because the visual noise increased and decreased in each half. $W_A$ was larger for increasing visual noise in the 1st as compared to the 2nd half, while $w_A$ was smaller for decreasing visual noise in the 1st as compared to the 2nd half (*Figure 3B, C*). These impressions were confirmed statistically in 2 (1st vs. flipped 2nd half) x 9 (bins) repeated measures ANOVAs (*Table 1*) showing a significant main effect of the 1st versus flipped 2nd half period for the sinusoidal (F(1, 24)=12.162, p=0.002, partial $\eta^2$ = 0.336) and the RW1 sequence (F(1, 32)=14.129, p<0.001, partial $\eta^2$ = 0.306). For the RW2 sequence, we observed a significant interaction (F(4.6, 82.9)=3.385, p=0.010, partial $\eta^2$ = 0.158), because the visual noise did not change monotonically within each half period. Instead, monotonic increases and decreases in visual noise alternated at nearly the double frequency in RW2 as compared to RW1. The asymmetry in the auditory weights' time course across the three experiments suggested that the visual noise in the past influenced observers' current visual uncertainty estimate resulting in smaller auditory weights for ascending visual noise and greater auditory weights for descending visual noise.

To further investigate the influence of past visual noise on observers' auditory weights, we estimated a regression model in which the relative auditory weights $w_A$ for each of the 20 bins were predicted by the visual STD in the current bin and the difference in STD between the current and the previous bin (see *Equation 2*). Indeed, both the current visual STD (p<0.001 for all three sequences; Sinusoid: t(24)=15.767, Cohen's d = 3.153; RW1: t(32) = 15.907, Cohen's d = 2.769; RW2: t(18) = 12.978, Cohen's d = 2.977, two sided one-sample t test against zero) and the difference in STD between the current and the previous bin (i.e. Sinusoid t(24) = −3.687, p=0.001, Cohen's d = −0.737; RW1 t(32) = −2.593, p=0.014, Cohen's d = −0.451; RW2 t(18) = -2.395, p=0.028,

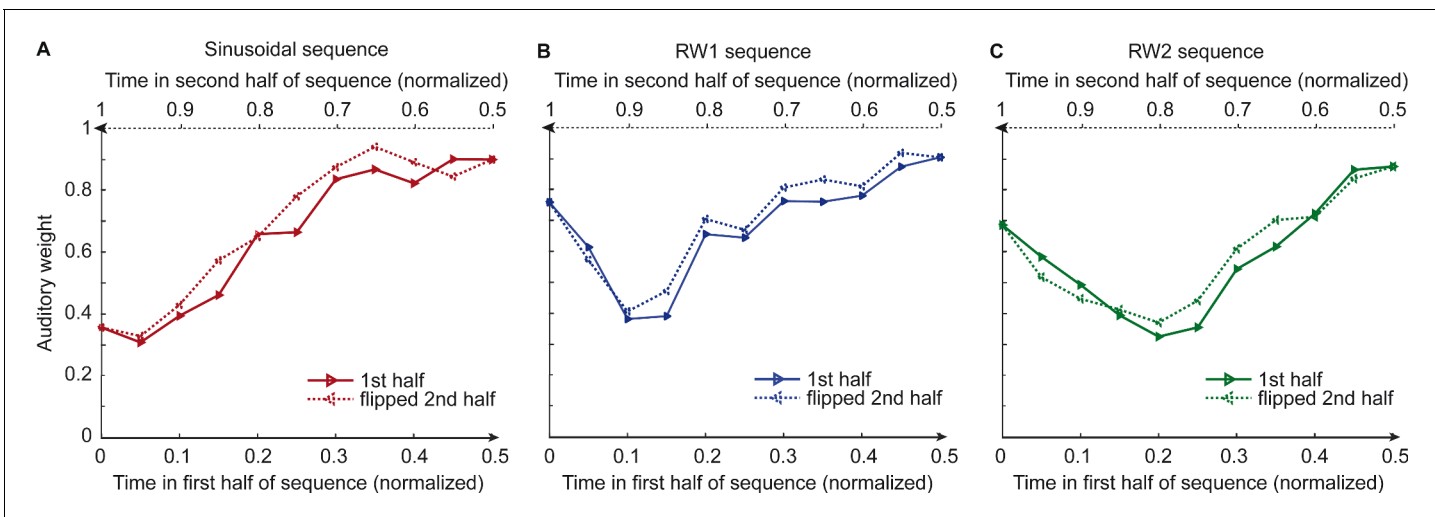

**Figure 3.** Observers' relative auditory weights for continuous sequences of visual noise. Relative auditory weights $w_A$ of the 1st (solid) and the flipped 2nd half (dashed) of a period (binned into 20 bins) plotted as a function of the normalized time in the sinusoidal (red), the RW1 (blue), and the RW2 (green) sequences. Relative auditory weights were computed from auditory localization responses of human observers.

**Table 1.** Analyses of the temporal asymmetry of the relative auditory weights across the four sequences of visual noise using repeated measures ANOVAs with the factors sequence part (1st vs. flipped 2nd half), bin and jump position (only for the sinusoidal sequences with intermittent jumps).

| | Effect | F | df1 | df2 | p | Partial $\eta^2$ |
|---|---|---|---|---|---|---|
| Sinusoid | Part | 12.162 | 1 | 24 | 0.002 | 0.336 |
| | Bin | 92.007 | 3.108 | 74.584 | <0.001 | 0.793 |
| | PartXBin | 2.167 | 2.942 | 70.617 | 0.101 | 0.083 |
| RW1 | Part | 14.129 | 1 | 32 | 0.001 | 0.306 |
| | Bin | 76.055 | 4.911 | 157.151 | <0.001 | 0.704 |
| | PartXBin | 1.225 | 4.874 | 155.971 | 0.300 | 0.037 |
| RW2 | Part | 2.884 | 1 | 18 | 0.107 | 0.138 |
| | Bin | 60.142 | 3.304 | 59.467 | <0.001 | 0.770 |
| | PartXBin | 3.385 | 4.603 | 82.849 | 0.010 | 0.158 |
| Sinusoid with intermittent jumps | Jump | 28.306 | 2 | 34 | <0.001 | 0.625 |
| | Part | 24.824 | 1 | 17 | <0.001 | 0.594 |
| | Bin | 76.476 | 1.873 | 31.839 | <0.001 | 0.818 |
| | JumpXPart | 0.300 | 2 | 34 | 0.743 | 0.017 |
| | JumpXBin | 8.383 | 3.309 | 56.247 | <0.001 | 0.330 |
| | PartXBin | 1.641 | 3.248 | 55.222 | 0.187 | 0.088 |
| | JumpXPartXBin | 0.640 | 5.716 | 97.175 | 0.690 | 0.036 |

Note: The factor bin comprised nine levels in the first three and seven levels in the fourth sequence. In this sequence, the factor Jump comprised three levels. If Mauchly tests indicated significant deviations from sphericity (p<0.05), we report Greenhouse-Geisser corrected degrees of freedom and p values.

Cohen's d = −0.549) significantly predicted observers' relative auditory weights (for complementary results of nested model comparisons see Appendix 1 and *Supplementary file 1-Table 5*). Collectively, these results suggest that observers' visual uncertainty estimates (as indexed by the relative auditory weights $w_A$) depend not only on the current sensory signal, but also on the recent history of the sensory noise. These results were also validated in a control analysis that regressed out and thus accounted for potential influences of the previous visual location on observers' sound localization, suggesting that the effects of past visual uncertainty cannot be explained by effects of past visual location mean (Appendix 1, *Figure 2—figure supplement 1*, *Supplementary file 1-tables 2-4*).

To characterize how human observers use information from the past to estimate current sensory uncertainty, we compared three computational models that differed in how visual uncertainty is learnt over time (*Figure 4*): Model 1, the instantaneous learner, estimates visual uncertainty independently for each trial as assumed by current standard models. Model 2, the optimal Bayesian learner, estimates visual uncertainty by updating the prior uncertainty estimate obtained from past visual signals with the uncertainty estimate from the current signal. Model 3, the exponential learner, estimates visual uncertainty by exponentially discounting past uncertainty estimates. All three models account for observers' uncertainty about whether auditory and visual signals were generated by common or independent sources by explicitly modeling the two potential causal structures (*Körding et al., 2007*) underlying the audiovisual signals (n.b. only the model component pertaining to the 'common cause' case is shown in *Figure 1B*, for the full model see *Figure 1—figure supplement 1*). Models were fit individually to observers' data by sampling from the posterior over parameters for each observer (*Table 2*).

We compared the three models in a fixed and random effects analysis (*Penny et al., 2010*; *Rigoux et al., 2014*) using the Watanabe-Akaike information criterion (WAIC) as appropriate for evaluating model samples (*Gelman et al., 2014*) (i.e. a low WAIC indicates a better model, a difference greater than 10 is considered very strong evidence for a model). In the fixed-effects analysis (see *Table 2* for details), the Bayesian learner was substantially better than the instantaneous learner across all three experiments, but outperformed the exponential learner reliably only in the sinusoidal sequence. Likewise, the random-effects analysis based on hierarchical Bayesian model selection

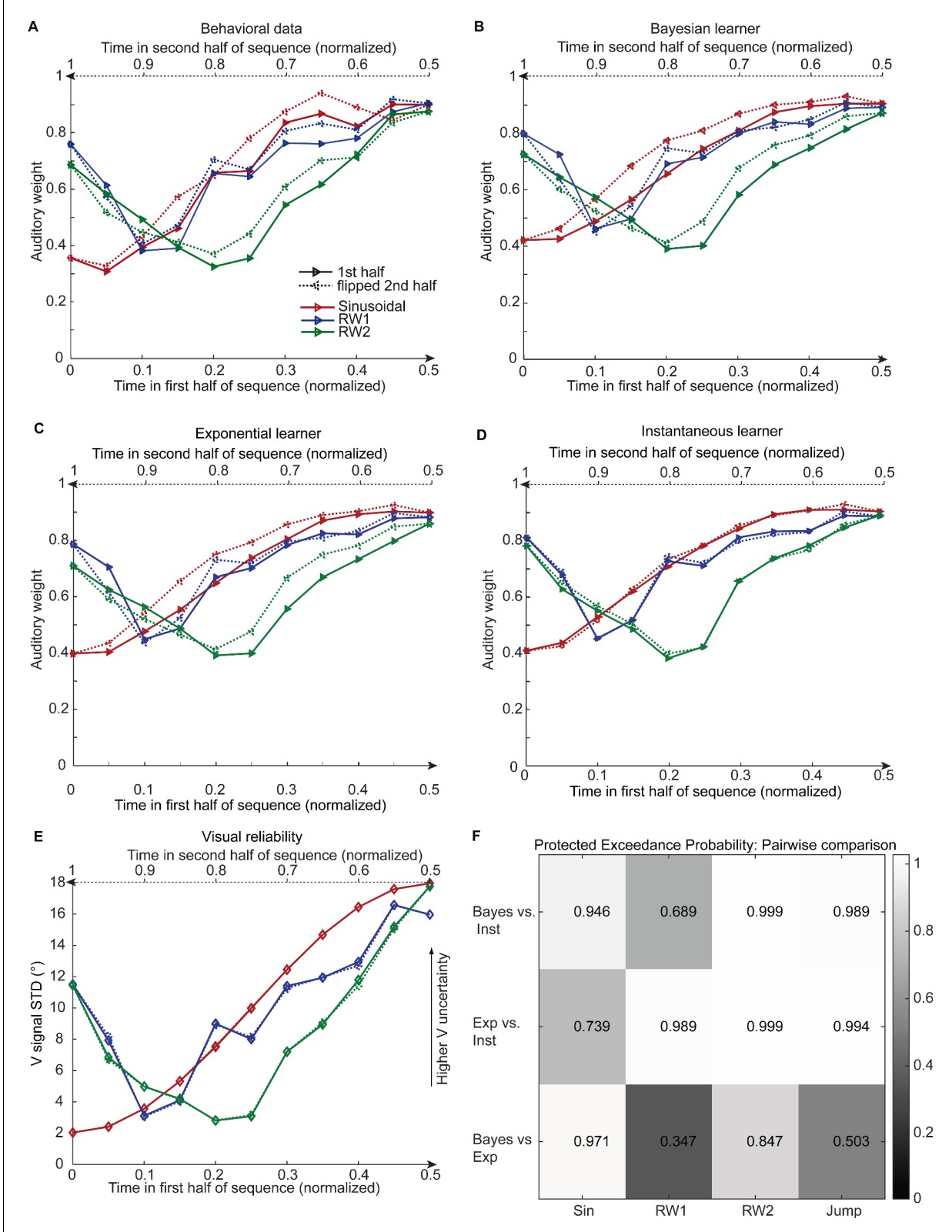

**Figure 4.** Observed and predicted relative auditory weights for continuous sequences of visual noise. Relative auditory weights $w_A$ of the 1st (solid) and the flipped 2nd half (dashed) of a period (binned into 20 bins) plotted as a function of the normalized time in the sinusoidal (red), the RW1 (blue) and the RW2 (green) sequences. Relative auditory weights were computed from auditory localization responses of human observers (**A**), Bayesian (**B**), exponential (**C**), or instantaneous (**D**) learning models. For comparison, the standard deviation of the visual signal is shown in (**E**). Please note that all

*Figure 4 continued on next page*

*Figure 4 continued*

models were fitted to observers' auditory localization responses (i.e. not the auditory weight $w_A$). (**F**) Bayesian model comparison – Random effects analysis: The matrix shows the protected exceedance probability (color coded and indicated by the numbers) for pairwise comparisons of the Instantaneous (Inst), Bayesian (Bayes) and Exponential (Exp) learners separately for each of the four experiments. Across all experiments we observed that the Bayesian or the Exponential learner outperformed the Instantaneous learner (i.e. a protected exceedance probability >0.94) indicating that observers used the past to estimate sensory uncertainty. However, it was not possible to arbitrate reliably between the Exponential and the Bayesian learner across all experiments (protected exceedance probability in bottom row).

(*Penny et al., 2010*; *Rigoux et al., 2014*) showed a protected exceedance probability that was substantially greater for the Bayesian learner (Sin, RW2) or the exponential learner (RW1, RW2) than for the instantaneous learner (*Figure 4F*). However, the direct comparison between the Bayesian and the exponential learner did not provide consistent results across experiments. As shown in *Figure 4A and B*, both the Bayesian and the exponential learner accurately reproduced the temporal asymmetry for the auditory weights across all three experiments.

From the optimal Bayesian learner, we inferred observers' estimated rate of change in visual reliability (i.e. parameter $\frac{1}{\kappa}$). The sinusoidal sequence was estimated to change at a faster pace (median $\kappa$ = 7.4 across observers, 95% confidence interval, 95% CI [4.8, 10.8] estimated via bootstrapping) than the RW1 sequence (median $\kappa$ = 8.1, 95% CI [7.0,14.9]), but slower than the RW2 sequence (median $\kappa$ = 6.7, 95% CI [4.4,11.2]) indicating that the Bayesian learner accurately inferred that visual reliability changed at different pace across the three continuous sequences (see legend of *Figure 2*). Likewise, the learning rates 1-$\gamma$ of the exponential learner accurately reflect the different rates of change across the sequences (Sinusoid $\gamma = 0.23$, 95% CI [0.14, 0.28]; RW1: $\gamma = 0.33$, 95% CI [0.21, 0.38]; RW2: $\gamma = 0.25$, 95% CI [0.21, 0.29]). Both the Bayesian and the exponential learner thus estimated a smaller rate of change for the RW1 than for the sinusoidal sequence – although caution needs to be applied when interpreting these results given the extensive confidence intervals. Further, the learning rates of the exponential learner imply that observers gave the visual signals presented 4.1 (Sinusoid), 5.4 (RW1), and 4.3 (RW2) seconds before the current stimulus 5% of the weight they assigned to the current visual signal to estimate the visual reliability.

To further disambiguate between the Bayesian and the exponential learner, we designed a fourth experimental 'jump sequence' that introduced abrupt increases or decreases in physical visual noise at three positions into the sinusoidal sequence (*Figure 5A*). Using the same analysis approach as for experiments 1–3, we replicated the temporal asymmetry for the auditory weights (*Figure 5B*). For all

**Table 2.** Model parameters (median), absolute WAIC and relative.
ΔWAIC values for the three candidate models in the four sequences of visual noise.

| Sequence | Model | $\sigma_A$ | $P_{common}$ | $\sigma_0$ | $\kappa$ or $\gamma$ | WAIC | ΔWAIC |
|---|---|---|---|---|---|---|---|
| Sinusoid | Instantaneous learner | 5.56 | 0.63 | 8.95 | - | 81931.2 | 109.9 |
| | Bayesian learner | 5.64 | 0.65 | 9.03 | $\kappa$: 7.37 | 81821.3 | 0 |
| | Exponential discounting | 5.62 | 0.64 | 9.02 | $\gamma$: 0.23 | 81866.9 | 45.6 |
| RW1 | Instantaneous learner | 6.30 | 0.69 | 8.46 | - | 110051.2 | 89.0 |
| | Bayesian learner | 6.29 | 0.72 | 8.68 | $\kappa$: 8.06 | 109962.2 | 0 |
| | Exponential discounting | 6.26 | 0.70 | 8.75 | $\gamma$: 0.33 | 109929.9 | −32.3 |
| RW2 | Instantaneous learner | 6.36 | 0.72 | 10.79 | - | 62576.4 | 201.3 |
| | Bayesian learner | 6.49 | 0.78 | 10.9 | $\kappa$: 6.7 | 62375.2 | 0 |
| | Exponential discounting | 6.46 | 0.73 | 11.0 | $\gamma$: 0.25 | 62421.5 | 46.3 |
| Sinusoid with intermittent jumps | Instantaneous learner | 6.38 | 0.65 | 8.19 | - | 83891.4 | 94.9 |
| | Bayesian learner | 6.45 | 0.68 | 8.26 | $\kappa$: 6.13 | 83796.5 | 0 |
| | Exponential discounting | 6.43 | 0.67 | 8.20 | $\gamma$: 0.24 | 83798.1 | 1.64 |

Note: WAIC values were computed for each participant and summed across participants. A low WAIC indicates a better model. ΔWAIC is relative to the WAIC of the Bayesian learner.

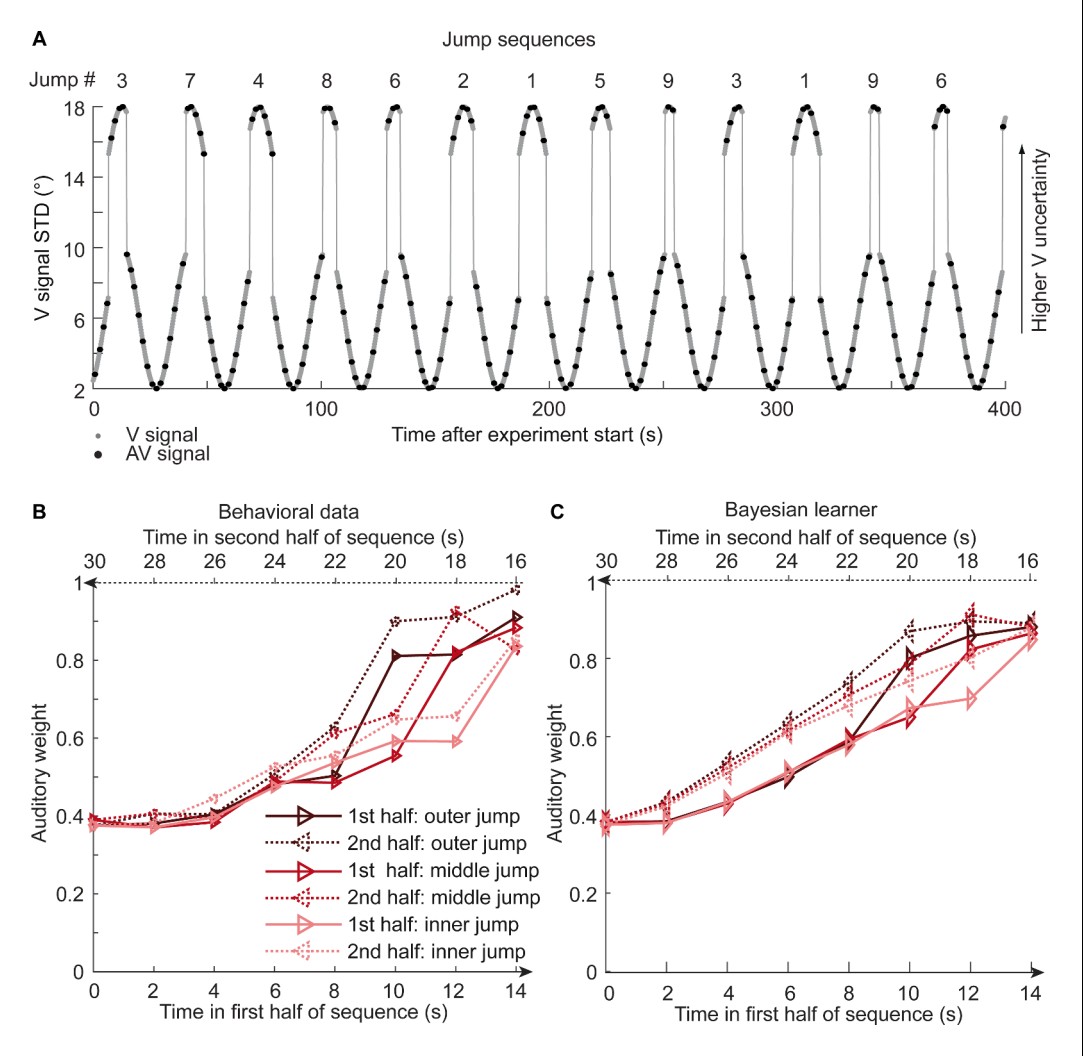

**Figure 5.** Time course of visual noise and relative auditory weights for sinusoidal sequence with intermittent jumps in visual noise (N = 18). (**A**) The visual noise (i.e. STD of the cloud of dots, right ordinate) is displayed as a function of time. Each cycle included one abrupt increase and decrease in visual noise. The sequence of visual clouds was presented every 200 ms (i.e. at 5 Hz) while audiovisual (AV) signals (black dots) were interspersed with an inter-trial asynchrony jittered between 1.4 and 2.8 s. (**B, C**) Relative auditory weights $w_A$ of the 1st (solid) and the flipped 2nd half (dashed) of a period (binned into 15 bins) plotted as a function of the time in the sinusoidal sequence with intermitted inner (light gray), middle (gray), and outer (dark gray) jumps. Relative auditory weights were computed from auditory localization responses of human observers (**B**) and the Bayesian learning model (**C**). Please note that all models were fitted to observers' auditory localization responses (i.e. not the auditory weight $w_A$).

The online version of this article includes the following figure supplement(s) for figure 5:

**Figure supplement 1.** Time course of relative auditory weights and visual noise for the sinusoidal sequence with intermittent jumps in visual noise for the exponential and instantaneous learning models.

**Figure supplement 2.** Time course of relative auditory weights and root mean squared error of the computational models before and after the jumps in the sinusoidal sequence with intermittent jumps.

three 'jump positions', $w_A$ was significantly smaller for the 1st half of each period, when visual noise increased, than the 2nd half, when visual noise decreased over time. The 3 (jump positions) x 2 (1st vs. flipped 2nd half) x 7 (bins) repeated measures ANOVA showed a significant main effect of 1st versus flipped 2nd period's half (F(1,17) = 24.824, p<0.001, partial $\eta^2$ = 0.594), while this factor was not involved in any higher-order interaction (see *Table 1*). Further, in a regression model the current visual STD (t(17) = 11.655, p<0.001, Cohen's d = 2.747) and the difference between current and

previous STD (t(17) = −4.768, p<0.001, Cohen's d = −1.124) significantly predicted the relative auditory weights. Thus, we replicated our finding that the visual noise in the past influenced observers' current visual uncertainty estimate as indexed by the relative auditory weights $w_A$.

Bayesian model comparison using a fixed-effects analysis showed that both the Bayesian learner and the exponential learner substantially outperformed the instantaneous learner (see *Table 2*). However, consistent with our Bayesian model comparison results for the continuous sequences, the Bayesian learner did not provide a better explanation for observers' responses than the exponential learner (ΔWAIC = +2, see *Table 2*, *Figure 5C* and *Figure 5—figure supplement 1A*). Likewise, a random-effects analysis based on hierarchical Bayesian model selection showed that the Bayesian and the exponential learners outperformed the instantaneous learner, but again we were not able to adjudicate between the Bayesian and exponential learner (*Figure 4F*, see also methods and results in Appendix 1, *Figure 5—figure supplement 2* and *Supplementary file 1-Table 6* for further analyses justifying the choice of continuous learning models in the jump sequence).

In summary, across four experiments that used continuous and discontinuous sequences of visual noise, we have shown that the Bayesian or exponential learners outperform the instantaneous learner. However, across the four experiments we were not able to decide whether observers adapted to changes in visual noise according to a Bayesian or an exponential learner. The key feature that distinguishes between the Bayesian and the exponential learner is that only the Bayesian learner adapts dynamically based on its uncertainty about its visual reliability estimates. As a consequence, the Bayesian learner should adapt faster than the exponential learner to increases in physical visual noise (i.e. spread of the visual cloud) but slower to decreases in visual noise. From the Bayesian learner's perspective, the faster learning for increases in visual noise emerges because it is unlikely that visual dots form a large spread cloud under the assumption that the true visual spread of the cloud is small. Conversely, the Bayesian learner will adapt more slowly to decreases in visual variance, because under the assumption of a visual cloud with a large spread visual dots may form a small cloud by chance. Indeed, previous research has shown that observers adapt their variance estimates faster for changes from small to large than for changes from large to small variance (*Berniker et al., 2010*). However, these results have been shown for learning about a hidden variable such as the prior that defines the spatial distribution from which an object's location is sampled. In our study, we manipulated the variance of the likelihood, that is the variance of the clouds of dots.

Asymmetric differences in adaptation rate between the exponential and the Bayesian learner should thus be amplified if we increase observer's uncertainty about its visual reliability estimate by reducing the number of dots of the visual cloud from 20 to 5 dots. Based on simulations, we therefore explored whether we could experimentally discriminate between the Bayesian and exponential learner using continuous sinusoidal or discontinuous 'jump' sequences with visual clouds of only five dots. For the two sequences, we simulated the sound localization responses of 12 observers based on the Bayesian learner model and fitted the Bayesian and exponential learner models to the responses of each simulated Bayesian observer. *Figure 6* shows observers' auditory weights indexing their estimated visual reliability across time that we obtained from the fitted responses of the Bayesian (blue) and the exponential learner (green). The simulations reveal the characteristic differences in how the Bayesian and the exponential learner adapt their visual uncertainty estimates to increases and decreases in visual noise. As expected, the Bayesian learner adapts its visual uncertainty estimates faster than the exponential learner to increases in visual noise, but slower to decreases in visual noise. Nevertheless, these differences are relatively small, so that the difference in mean log likelihood between the Bayesian and exponential learner is only −1.82 for the sinusoidal sequence and −2.74 for the jump sequence.

Next, we investigated whether our experiments successfully mimicked situations in which observers benefit from integrating past and current information to estimate their sensory uncertainty. We compared the accuracy of the instantaneous, exponential and Bayesian learner's visual uncertainty estimates in terms of their mean absolute deviation (in percentage) from the true variance. For Gaussian clouds of 20 dots, the instantaneous learner's error in the visual uncertainty estimates of 21.7% is reduced to 13.7% and 14.9% for the exponential and Bayesian learners, respectively (with best fitted γ = 0.6, in the sinusoidal sequence). For Gaussian clouds composed of only five dots, the exponential and Bayesian learners even cut down the error by half (i.e. 46.8% instantaneous learner, 29.5% exponential learner, 23.9% Bayesian learner, with best fitted γ = 0.7).

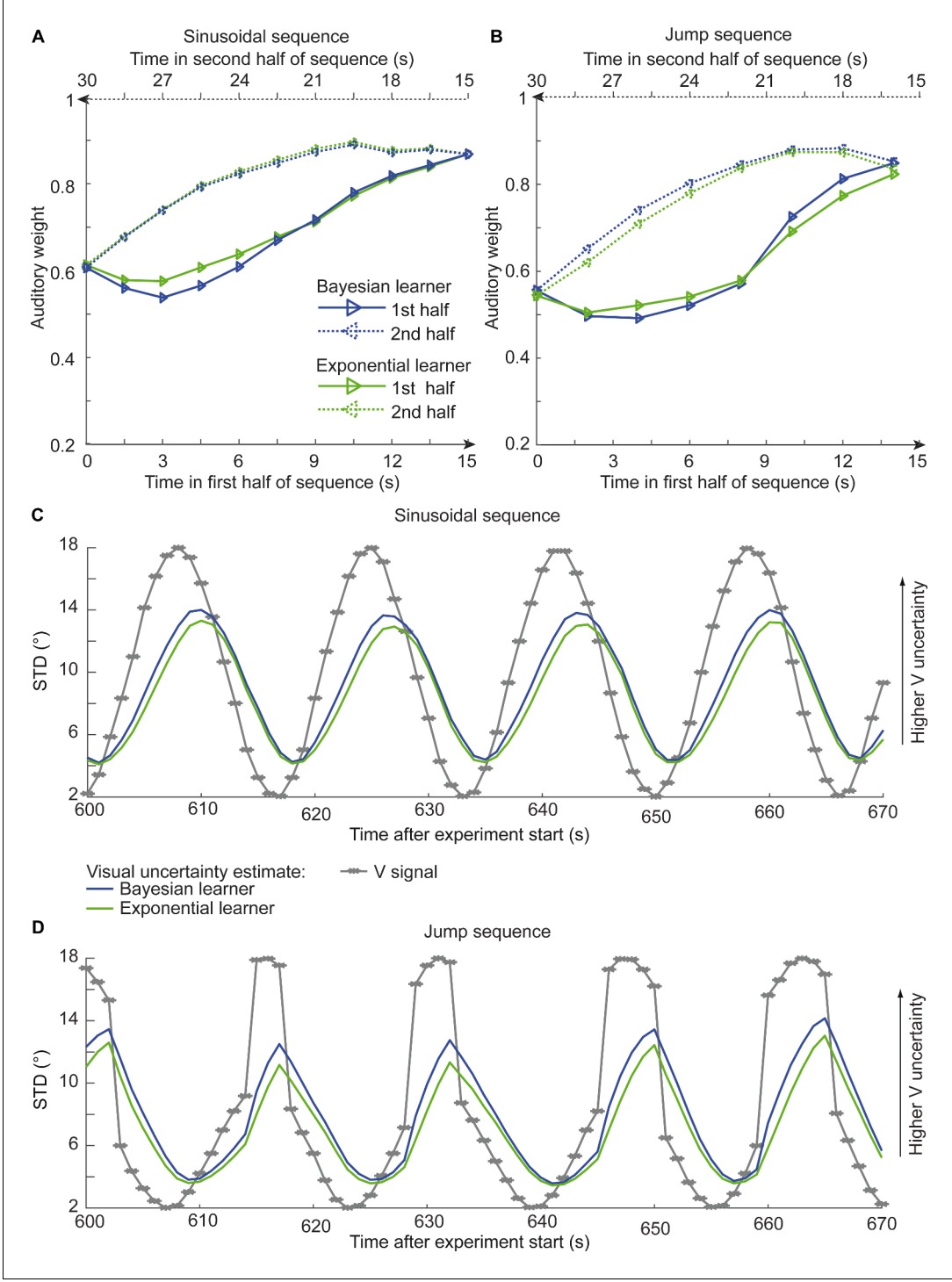

**Figure 6.** Time course of the relative auditory weights, the standard deviation (STD) of the visual cloud and the STD of the visual uncertainty estimates. (A) Relative auditory weights w_A of the 1st (solid) and the flipped 2nd half (dashed) of a period (binned into 15 bins) plotted as a function of the time in the sinusoidal sequence. Relative auditory weights were computed from the predicted auditory localization responses of the Bayesian (blue) or exponential (green) learning models fitted to the simulated localization responses of a Bayesian learner based on visual clouds of 5 dots. (B) Relative auditory weights w_A computed as in (A) for the sinusoidal sequence with intermitted jumps. Only the outer-most jump (dark brown in *Figure 5B/C* and *Figure 5—figure supplement 1*) is shown. (C, D) STD of the visual cloud of 5 dots (gray) and the STD of observers' visual uncertainty as estimated by the Bayesian (blue) and exponential (green) learners (that were fitted to the simulated localization responses of a

*Figure 6 continued on next page*

*Figure 6 continued*

Bayesian learner) as a function of time for the sinusoidal sequence (C) and in the sinusoidal sequence with intermitted jumps (D). Note that only an exemplary time course from 600 to 670 s after the experiment start is shown.

Collectively, these simulation results suggest that even in situations in which observers benefit from combining past with current sensory inputs to obtain more precise uncertainty estimates, the exponential learner is a good approximation of the Bayesian learner, making it challenging to dissociate the two experimentally based on noisy human behavioral responses.

## Discussion

The results from our four experiments challenge classical models of perceptual inference where a perceptual interpretation is obtained using a likelihood that depends solely on the current sensory inputs (*Ernst and Banks, 2002*). These models implicitly assume that sensory uncertainty (i.e. likelihood variance) is instantaneously and independently accessed from the sensory signals on each trial based on initial calibration of the nervous system (*Jacobs and Fine, 1999*). Most prominently, in the field of cue combination it is generally assumed that sensory signals are weighted by their uncertainties that are estimated only from the current sensory signals (*Alais and Burr, 2004*; *Ernst and Banks, 2002*; *Jacobs, 1999*) (but see *Mikula et al., 2018*; *Triesch et al., 2002*).

By contrast, our results demonstrate that human observers integrate inputs weighted by uncertainties that are estimated jointly from past and current sensory signals. Across the three continuous and the one discontinuous jump sequences, observers' current visual reliability estimates were influenced by visual inputs that were presented 4–5 s in the past albeit their influence amounted to only 5% of the current visual signals.

Critically, observers adapted their visual uncertainty estimates flexibly according to the rate of change in the visual noise across the experiments. As predicted by both Bayesian and exponential learning models, observers' visual reliability estimates relied more strongly on past sensory inputs, when the visual noise changed more slowly across time. While observers did not explicitly notice that each of the four experiments was composed of repetitions of temporally symmetric sequence components, we cannot fully exclude that observers may have implicitly learnt this underlying temporal structure. However, implicit or explicit knowledge of this repetitive sequence structure should have given observers the ability to predict and preempt future changes in visual reliability and therefore attenuated the temporal lag of the visual reliability estimates. Put differently, our experimental choice of repeating the same sequence component over and over again in the experiment cannot explain the influence of past signals on observers' current reliability estimate, but should have reduced or even abolished it.

Importantly, the key feature that distinguishes the Bayesian from the exponential learner is how the two learners adapt to increases versus decreases in visual noise. Only the Bayesian learner represents and accounts for its uncertainty about its visual reliability estimates. As compared to the exponential learner, it should therefore adapt faster to increases but slower to decreases in visual noise (e.g. see *Berniker et al., 2010*). Our simulation results show this profile qualitatively, when the learner's uncertainty about its visual reliability estimate is increased by reducing the number of dots (see *Figure 6*). But even for visual clouds of five dots, the differences in learning curves between the Bayesian and exponential learner are very small making it difficult to adjudicate between them given noisy observations from real observers. Unsurprisingly, therefore, Bayesian model comparison showed consistently across all four experiments that observers' localization responses can be explained equally well by an optimal Bayesian and an exponential learner. These results converge with a recent study showing that learning about a hidden variable such as observers' priors can be accounted for by an exponential averaging model (*Norton et al., 2019*).

Collectively, our experimental and simulation results suggest that under circumstances where observers substantially benefit from combining past and current sensory inputs for estimating sensory uncertainty, optimal Bayesian learning can be approximated well by more simple heuristic strategies of exponential discounting that update sensory weights with a fixed learning rate irrespective of observers' uncertainty about their visual reliability estimate (*Ma and Jazayeri, 2014*; *Shen and*

*Ma, 2016*). Future research will need to assess whether observers adapt their visual uncertainty estimates similarly if visual noise is manipulated via other methods such as stimulus luminance, duration, or blur.

From the perspective of neural coding, our findings suggest that current theories of probabilistic population coding (*Beck et al., 2008*; *Ma et al., 2006*; *Hou et al., 2019*) may need to be extended to accommodate additional influences of past experiences on neural representations of sensory uncertainties. Alternatively, the brain may compute sensory uncertainty using strategies of temporal sampling (*Fiser et al., 2010*).

In conclusion, our study demonstrates that human observers do not access sensory uncertainty instantaneously from the current sensory signals alone, but learn sensory uncertainty over time by combining past experiences and current sensory inputs as predicted by an optimal Bayesian learner or approximate strategies of exponential discounting. This influence of past signals on current sensory uncertainty estimates is likely to affect learning not only at slower timescales across trials (i.e. as shown in this study), but also at faster timescales of evidence accumulation within a trial (*Drugowitsch et al., 2014*). While our research unravels the impact of prior sensory inputs on uncertainty estimation in a cue combination context, we expect that they reveal fundamental principles of how the human brain computes and encodes sensory uncertainty.

## Materials and methods

### Participants

Seventy-six healthy volunteers participated in the study after giving written informed consent (40 female, mean age 25.3 years, range 18–52 years). All participants were naïve to the purpose of the study. All participants had normal or corrected-to normal vision and reported normal hearing. The study was approved by the human research review committee of the University of Tuebingen (approval number 432 2007 BO1) and the research review committee of the University of Birmingham (approval number ERN_11–0470P).

### Stimuli

The visual spatial stimulus was a Gaussian cloud of twenty bright gray dots (0.56° diameter, vertical STD 1.5°, luminance 106 cd/m$^2$) presented on a dark gray background (luminance 62 cd/m$^2$, i.e. 71% contrast). The auditory spatial cue was a burst of white noise with a 5 ms on/off ramp. To create a virtual auditory spatial cue, the noise was convolved with spatially specific head-related transfer functions (HRTFs). The HRTFs were pseudo-individualized by matching participants' head width, heights, depth, and circumference to the anthropometry of subjects in the CIPIC database (*Algazi et al., 2001*). HRTFs from the available locations in the database were interpolated to the desired locations of the auditory cue.

### Experimental design and procedure

In a spatial ventriloquist paradigm, participants were presented with audiovisual spatial signals. Participants indicated the location of the sound by pressing one of five spatially corresponding buttons and were instructed to ignore the visual signal. Participants did not receive any feedback on their localization response. The visual signal was a cloud of 20 dots sampled from a Gaussian. The visual clouds were re-displayed with variable horizontal STDs (see below) every 200 ms (i.e. at a rate of 5 Hz; *Figure 1A*). The cloud's location mean was temporally independently resampled from five possible locations (−10°, −5°, 0°, 5°, 10°) on each trial with the inter-trial asynchrony jittered between 1.4 and 2.8 s in steps of 200 ms. In synchrony with the change in the cloud's location, the dots changed their color and a concurrent sound was presented. The location of the sound was sampled from ±5° visual angle with respect to the mean of the visual cloud. Observers' visual uncertainty estimate was quantified in terms of the relative weight of the auditory signal on the perceived sound location. The change in the dot's color and the emission of the sound occurred in synchrony to enhance audiovisual binding.

## Continuous sinusoidal and RW sequences

Critically, to manipulate visual noise over time, the cloud's STD changed at a rate of 5 Hz according to (i) a sinusoidal sequence, (ii) an RW sequence 1 or (iii) an RW sequence 2 (*Figure 2*). In all sequences, the horizontal STD of the visual cloud spanned a range from 2 to 18°:

i.  *Experiment1 - Sinusoidal sequence (Sinusoid):* A sinusoidal sequence was generated with a period of 30 s. During the ~65 min of the experiment, each participant completed ~130 cycles of the sinusoidal sequence.
ii. *Experiment2 - Random walk sequence 1 (RW1):* First, we generated an RW sequence of 60 s duration using a Markov chain with 76 discrete states and transition probabilities of stay (1/3), change to lower (1/3) or upper (1/3) adjacent states. To ensure that the RW sequence segment starts and ends with the same value, this initial 60-s sequence segment was concatenated with its temporally reversed segment resulting in an RW sequence segment of 120 s duration. Each participant was presented with this 120 s RW1 sequence approximately 32 times during the experiment.
iii. *Experiment3 - Random walk sequence 2 (RW2):* Likewise, we created a second random-walk sequence of 15 s duration using a Markov chain with only 38 possible states and transition probabilities similar to above. The 15-s sequence was concatenated with its temporally reversed version resulting in a 30-s sequence. The smoothness of this sequence segment was increased by filtering it (without phase shift) with a moving average of 250 ms. Each participant was presented with this sequence segment ~130 times.

Generally, a session of a sinusoid, RW1, or RW2 sequence included 1676 trials. Because of experimental problems, four sessions included only 1128, 1143, or 1295 trials. Before the experimental trials, participants practiced the auditory localization task in 25 unimodal auditory trials, 25 audiovisual congruent trials with a single dot as visual spatial cue and 75 trials with stimuli as in the main experiment.

## Experiment 4 - Sinusoidal sequence with intermittent changes in visual noise (sinusoidal jump sequence)

To dissociate the Bayesian learner from approximate exponential discounting, we designed a sinusoidal sequence (period = 30 s) with intermittent increases/decreases in visual variance (*Figure 5*). As shown in *Figure 5A*, we inserted increases by 8° in visual STD at three levels of visual STD: 7.2°, 8.6°, 9.6° STD. Conversely, we inserted decreases by 8° in visual STD at 15.3°, 16.7°, 17.7° STD. We inserted jumps selectively in the period sections of high visual variance to make the jumps less apparent and maximize the chances that observers treated the series as a continuous sequence. As a result, the up-jumps occurred when the increases in visual variance were fastest (i.e. steeper slope), while the down-jumps occurred after sections in which the visual variance was relatively constant (i.e. shallow slope). We factorially combined these 3 (increases) x 3 (decreases) such that each sinewave cycle included exactly one sudden increase and decrease in visual STD (i.e. nine jump types). Otherwise, the experimental paradigm and stimuli were identical to the continuous sinusoidal sequence described above. During the ~80 min of this experiment, each participant completed ~154 cycles of the sinusoidal sequence including 16–18 cycles for each of the nine jump types. This sinusoidal jump sequence was expected to maximize differences in adaptation rate for the Bayesian and exponential learner. If participants continuously update their estimates of the visual reliability, as opposed to using a change point model (*Adams and Mackay, 2007*; *Heilbron and Meyniel, 2019*), the exponential learner will weight past and present uncertainty estimates throughout the entire sequence according to the same exponential function. By contrast, the Bayesian learner will take into account its uncertainty about the visual reliability and therefore adapt its visual reliability estimate for jumps from high to low visual variance (resp. low to high visual reliability, see *Figure 6*) more slowly than the exponential learner (see Appendix 1).

### Subject numbers and inclusion criteria

Of the 76 subjects, 30 participated in the sinusoidal and the RW1 sequence session. Eight additional subjects participated only in the RW1 sequence session. Eighteen additional subjects participated in the RW2 sequence session. One participant completed all three continuous sequences. Twenty subjects participated in the sinusoidal sequence with intermittent changes in visual uncertainty. In total,

we collected data from 30 participants for the sinusoidal, 38 participants for the RW1, 19 participants for the RW2, and 20 participants for the sinusoidal jump sequence. The sample sizes of 20–38 participants were based on a pilot experiment, which showed individually significant effects of past visual noise on the weighting of audiovisual spatial signals in 6/6 pilot participants. From these samples, we excluded participants if their perceived sound location did not depend on the current visual reliability (i.e. inclusion criterion p<0.05 in the linear regression; please note that this inclusion criterion is orthogonal to the question of whether participants' visual uncertainty estimate depends on visual signals prior to the current trial). Thus, we excluded five participants of the sinusoidal and RW1 sequence and two participants from the sinusoidal jump sequence. Finally, we analyzed data from 25 participants for the sinusoidal, 33 participants for the RW1, 19 participants for the RW2, and 18 participants for the sinusoidal jump sequence.

## Experimental setup

Audiovisual stimuli were presented using Psychtoolbox 3.09 (*Brainard, 1997*; *Kleiner et al., 2007*) (http://www.psychtoolbox.org) running under Matlab R2010b (MathWorks) on a Windows machine (Microsoft XP 2002 SP2). Auditory stimuli were presented at ~75 dB SPL using headphones (Sennheiser HD 555). As visual stimuli required a large field of view, they were presented on a 30″ LCD display (Dell UltraSharp 3007WFP). Participants were seated at a desk in front of the screen in a darkened booth, resting their head on an adjustable chin rest. The viewing distance was 27.5 cm. This setup resulted in a visual field of approximately 100°. Participants responded via a standard QWERTY keyboard. Participants used the buttons [i, 9, 0, -, = ] with their right hand for localization responses.

## Data analysis
### Continuous sinusoidal and RW sequences

At trial onset the visual cloud's location mean was independently resampled from five possible locations ($-10°$, $-5°$, $0°$, $5°$, $10°$). Concurrently, the cloud's dots changed their color and a sound was presented sampled from $\pm5°$ visual angle with respect to the mean of the visual cloud. The inter-trial asynchrony was jittered between 1.4 and 2.8 s in steps of 200 ms. Therefore, across the experiment the trial onsets occurred at different times relative to the period of the changing visual cloud's STD resulting in a greater effective sampling rate than provided if the inter-trial asynchrony had been fixed.

For each period of the three continuous sinusoidal and RW sequences, we sorted the trials (i.e. trial-specific visual cloud's STD, visual location, auditory location, and observers' sound localization responses) into 20 temporally adjacent bins that covered one complete period of the changing visual STD. This resulted in about 1676 trials in total/20 bins = approximately 80 trials on average per bin in each subject (more specifically: a range of 52–96 (Sin), 52–92 (RW 1), or 71–93 (RW2) trials, for details see *Supplementary file 1-Table 1*).

We quantified the influence of the auditory and visual locations on observers' perceived auditory location for each bin by estimating a regression model separately for each bin (i.e. one regression model per bin). For instance, for bin = 1 we computed:

$$R_{A,trial,bin=1} = L_{A,trial,bin=1}\beta_{A,bin=1} + L_{V,trial,bin=1}\beta_{V,bin=1} + \beta_{const,bin=1} + e_{trial,bin=1} \qquad (1)$$

with $R_{A,trial,bin=1}$ = Localization response for trial t and bin 1; $L_{A,trial,bin=1}$ or $L_{V,trial,bin=1}$ = 'true' auditory or visual location for trial t and bin 1; $\beta_{A,bin=1}$ or $\beta_{V,bin=1}$ = auditory or visual weight for bin 1; $\beta_{const,bin=1}$ = constant term; $e_{trial,bin=1}$ = error term for trial t and bin 1. For each bin b, we thus obtained one auditory and one visual weight estimate. The *relative* auditory weight for a particular bin was computed as $w_{A,bin} = \beta_{A,bin} / (\beta_{A,bin} + \beta_{V,bin})$ (*Figure 2A–C*).

By design, the temporal evolution of the physical visual variance (i.e. STD of the visual cloud) is symmetric for each period in the sinusoidal, RW1 and RW2 sequences. In other words, for physical visual noise, the 1st half and the flipped 2nd half within a period are identical (*Figure 3E*). Given this symmetry constraint, we evaluated the influence of past visual noise on participants' auditory weight $w_{A,bin}$ by comparing the $w_A$ for the bins in the 1st half and the flipped 2nd half in a repeated measures ANOVA. If human observers estimate visual uncertainty by combining prior with current visual uncertainty estimates as expected for a Bayesian learner, $w_A$ should differ between the 1st half and

the mirror-symmetric flipped 2nd half of the sequence. More specifically, $w_A$ should be smaller for the 1st half in which visual variance increased than for the mirror-symmetric time points of the 2nd half in which visual variance decreased. To test this prediction, we entered the subject-specific $w_{A,bin}$ into 2 (1st vs. flipped 2nd half) x 9 (bins, i.e. removing the bins at maximal and minimal visual noise values) repeated measures ANOVAs separately for the sinusoidal, RW1 and RW2 experiments (*Table 1*). For the sinusoidal sequence, we expected a main effect of 'half' because the sequence increased/decreased monotonically within each half period. For the RW1 and RW2 sequences, an influence of prior visual noise might also be reflected in an interaction effect of 'half x bin' because these sequences increased/decreased non-monotonically within each half period.

To further test whether the noise of past visual signals influenced observers' current visual uncertainty estimate, we employed a regression model in which the relative auditory weights $w_{A,bin}$ were predicted by the visual STD in the current bin and the difference in STD between the current and the previous bin:

$$w_{A,bin} = \sigma_{V,bin}\, \text{ß}_{\sigma V} + \left(\sigma_{V,bin} - \sigma_{V,bin-1}\right) \text{ß}_{\Delta\sigma V} + \text{ß}_{const} + e_{bin} \qquad (2)$$

with $w_{A,bin}$ = relative auditory weight in bin b; $\sigma_{V,bin}$ = mean visual STD in current bin b or previous bin b-1; $\text{ß}_{const}$ = constant term; $e_{bin}$ = error term. To allow for generalization to the population level, the parameter estimates ($\text{ß}_{\sigma V}$, $\text{ß}_{\Delta\sigma V}$) for each participant were entered into two-sided one-sample t-tests at the between-subject random-effects level.

## Sinusoidal sequence with intermittent changes in visual uncertainty

For each period of the sinusoidal sequence with intermittent changes, we sorted the values for the physical visual cloud's variance (i.e. the cloud's STD) and sound localization responses into 15 temporally adjacent bins which were positioned to capture the jumps in visual noise. For analysis of these sequences, we recombined the first and second halves of the 3 (increases at low, middle, high) x 3 (decreases at low, middle, high) sinewave cycles into three types of sinewave cycles such that both jumps were at low (=outer jump), middle (=middle jump), or high (=inner jump) visual noise. This recombination makes the simplifying assumption that the jump position of the first half will have negligible effects on participants' uncertainty estimates of the second half. As a result of this recombination, each bin comprised at least 44–51 trials across participants (*Supplementary file 1-Table 1*). As for the continuous sequences, we quantified the auditory and visual influence on the perceived auditory location for each bin based on separate regression models for the 15 temporally adjacent bins (see *Equation 1*). Next, we independently computed the relative auditory weight $w_{A,bin}$ = $\text{ß}_{A,bin}$ / ($\text{ß}_{A,bin}$ + $\text{ß}_{V,bin}$) for each of the 15 temporally adjacent bins. We statistically evaluated the influence of past visual noise on participants' auditory weight on the $w_A$ in terms of the difference between 1st half and flipped 2nd half using a 2 (1st vs. flipped 2nd half) x 7 (bins) x 3 (jump: inner, middle, outer) repeated measures ANOVAs (*Table 1*).

## Computational models (for continuous and discontinuous sequences)

To further characterize whether and how human observers use their uncertainty about previous visual signals to estimate their uncertainty of the current visual signal, we defined and compared three models in which visual reliability ($\lambda_V$) was (1) estimated instantaneously for each trial (i.e. instantaneous learner), was updated via (2) Bayesian learning or (3) exponential discounting (i.e. exponential learner) (*Figure 1—figure supplement 1*).

In the following, we will first describe the generative model that accounts for the fact that (1) visual uncertainty usually changes slowly across trials (i.e. time-dependent uncertainty changes) and (2) auditory and visual signals can be generated by one common or two independent sources (i.e. causal structure). Using this generative model as a departure point, we then describe how the instantaneous learner, the Bayesian learner and the exponential learner perform inference. Finally, we will explain how we account for participants' internal noise and predict participants' responses from each model (i.e. the experimenter's uncertainty).

## Generative model

On each trial *t*, the subject is presented with an auditory signal $A_t$, from a source $S_{A,t}$ (see *Figure 1—figure supplement 1*) together with a visual cloud of dots at time *t* arising from a source, $S_{V,t}$, drawn

from a Normal distribution $S_{V,t} \sim N(0, 1/\lambda_S)$ with the spatial reliability (i.e. inverse of the spatial variance): $\lambda_S = 1/\sigma_S^2$. Critically, $S_{A,t}$ and $S_{V,t}$, can either be two independent sources (C = 2) or one common source (C = 1): $S_{A,t} = S_{V,t} = S_t$ (**Körding et al., 2007**).

We assume that the auditory signal is corrupted by noise, so that the internal signal is $A_t \sim N(S_{A,t}, 1/\lambda_A)$. By contrast, the individual visual dots (presented at high visual contrast) are assumed to be uncorrupted by noise, but presented dispersed around the location $S_{V,t}$ according to $V_{i,t} \sim N(U_t, 1/\lambda_{V,t})$, where $U_t \sim N(S_{V,t}, 1/\lambda_{V,t})$. The dispersion of the individual dots, $1/\lambda_{V,t}$, is assumed to be identical to the uncertainty about the visual mean, allowing subjects to use the dispersion as an estimate of the uncertainty about the visual mean.

The visual reliability of the visual cloud, $\lambda_{V,t} = 1/\sigma_{V,t}^2$, varies slowly at the re-display rate of 5 Hz according to a log RW: $\log \lambda_{V,t} \sim N\left(\log \lambda_{V,t-1}, 1/\kappa\right)$ with $\frac{1}{\kappa}$ being the variability of $\lambda_{V,t}$ in log space. We also use this log RW model to approximate learning in the four jump sequence (see **Behrens et al., 2007**).

The generative models of the instantaneous, Bayesian, and exponential learners all account for the causal uncertainty by explicitly modeling the two potential causal structures. Yet, they differ in how they estimate the visual uncertainty on each trial, which we will describe in greater detail below.

## Observer inference

The instantaneous, Bayesian, and exponential learners invert this (or slightly modified, see below) generative model during perceptual inference to compute the posterior probability of the auditory location, $S_{A,t}$, given the observed $A_t$ and $V_{i,t}$. The observer selects a response based on the posterior using a subjective utility function which we assume to be the minimization of the squared error $(S_{A,t} - S_{true})^2$. For all models, the estimate for the location of the auditory source is obtained by averaging the auditory estimates under the assumption of common and independent sources by their respective posterior probabilities (i.e. model averaging, see **Figure 1—figure supplement 1**):

$$\hat{S}_{A,t} = \hat{S}_{A,C=1,t}\, P\left(C_t = 1 | A_t, V_{1:n,t}\right) + \hat{S}_{A,C=2,t}\left(1 - P\left(C_t = 1 | A_t, V_{1:n,t}\right)\right) \tag{3}$$

where $\hat{S}_{A,C=1,t}$ and $\hat{S}_{A,C=2,t}$ depend on the model (see below), and $P\left(C = 1 | A_t, V_{1:n,t}\right)$ is the posterior probability that the audio and visual stimuli originated from the same source according to Bayesian causal inference (**Körding et al., 2007**).

$$P\left(C_t = 1 | A_t, V_{1:n,t}\right) = \frac{\left(P\left(A_t, V_{1:n,t} | C=1\right)P(C_t=1)\right)}{\left(P\left(A_t, V_{1:n,t} | C_t=1\right)P(C_t=1)\right) + \left(P\left(A_t, V_{1:n,t} | C_t=2\right)\left(1 - P(C_t=1)\right)\right)} \tag{4}$$

Finally, for all models, we assume that the observer pushes the button associated with the position closest to $\hat{S}_{A,t}$. In the following, we describe the generative and inference models for the instantaneous, Bayesian, and exponential learners. For the Bayesian learner, we focus selectively on the model component that assumes a common cause, C = 1 (for full derivation including both model components, see Appendix 2).

## Model 1: Instantaneous learner

The instantaneous learning model ignores that the visual reliability (i.e. the inverse of visual uncertainty) of the current trial depends on the reliability of the previous trial. Instead, it estimates the visual reliability independently for each trial from the spread of the cloud of visual dots:

$$
\begin{aligned}
P\left(S_{A,t}, U_t, \lambda_{V,t} | A_{1:t},\, V_{1:n,1:t}\right) &= P\left(S_{A,t}, U_t, \lambda_{V,t} | A_t,\, V_{1:n,t}\right) = \\
P\left(C=1 | A_t,\, V_{1:n,t}\right) &P_{C=1}\left(S_t, U_t, \lambda_{V,t} | A_t,\, V_{1:n,t}\right) + \\
P\left(C=2 | A_t,\, V_{1:n,t}\right) &P_{C=2}\left(S_{A,t}, U_t, \lambda_{V,t} | A_t,\, V_{1:n,t}\right) = \\
P\left(C=1 | A_t,\, V_{1:n,t}\right) &\frac{P(S_t)P(A_t|S_t)P_{C=1}\left(U_t|S_t, \lambda_{V,t}\right)P\left(V_{1:n,t}|U_t, \lambda_{V,t}\right)P\left(\lambda_{V,t}\right)}{Z_1} + \\
\left(1 - P\left(C=1 | A_t,\, V_{1:n,t}\right)\right) &P\left(S_{A,t}\right)P\left(A_t|S_{A,t}\right)/Z_2.
\end{aligned}
\tag{5}
$$

with $Z_1$, $Z_2$ as normalization constants.

Apart from $P(C=1|A_t, V_t)$, these terms are all normal distributions, while we assume in this model that $P\left(\lambda_{V,t}\right)$ is uninformative. Hence, visual reliability is computed from the variance:

$\widehat{\lambda_{Vt}} = 1/\left(\sigma^2_{Vt} + \frac{\sigma^2_{Vt}}{n}\right)$ where $\sigma^2_{Vt} = 1/(n-1)\sum_{i=1}^{n}(V_{i,t} - \bar{V}_{i,t})^2$ is the sample variance (and $\bar{V}_t = 1/n\sum_{i=1}^{n}V_{i,t}$ is the sample mean). The causal component estimates are given by:

$$\hat{S}_{A,C=1,t} = \frac{\hat{\lambda}_{V,t}\bar{V}_t + \lambda_A A_t}{\lambda_{V,t} + \lambda_A + \lambda_S} \tag{6}$$

$$\hat{S}_{A,C=2,t} = \frac{\lambda_A A_t}{\lambda_A + \lambda_S} \tag{7}$$

These two components are then combined based on the posterior probabilities of common and independent cause models (see **Equation 3**). This model is functionally equivalent to a Bayesian causal inference model as described in **Körding et al., 2007**, but with visual reliability computed directly from the sample variance rather than a fixed unknown parameter (which the experimenter estimates during model fitting).

## Model 2: Bayesian learner

The Bayesian learner capitalizes on the slow changes in visual reliability across trials and combines past and current inputs to provide a more reliable estimate of visual reliability and hence auditory location. It computes the posterior probability based on all auditory and visual signals presented until time t (here only shown for C = 1, see Appendix 2).

According to Bayes rule, the joint probability of all variables until time $t$ can be written based on the generative model as:

$$P(\lambda_{V,1:t}, A_{1:t}, U_{1:n,1:t}, V_{1:n,1:t}, S_{1:t}) =$$
$$P(A_1|S_1)P(V_{1:n,1}|U_1,\lambda_{V,1})P(U_1|S_1,\lambda_{V,1})P(S_1)P(\lambda_{V,1})$$
$$\prod_{k=2}^{t} P(A_k|S_k)P(V_{1:n,k}|U_k,\lambda_{V,k})P(U_k|S_k,\lambda_{V,k})P(\lambda_{V,k}|\lambda_{V,k-1})P(S_k) \tag{8}$$

As above, the visual likelihood is given by the product of individual Normal distributions for each dot $i$: $P(V_{1:n,t}|U_t,\lambda_{V,t}) = \prod_{i=1}^{n} N(V_{i,t}|U_t, 1/\lambda_{V,t})$, and $P(U_t|S_t,\lambda_{V,t}) = N(U_t|S_t, 1/\lambda_{V,t})$.

The prior $P(S_t)$ is a Normal distribution $N(S_t|0, 1/\lambda_S)$ and the auditory likelihood.

$P(A_t|S_t)$ is a Normal distribution $N(A_t|S_t, 1/\lambda_A)$. As described in the generative model, $P(\lambda_{V,k}|\lambda_{V,k-1})$ is given by $\log\lambda_{V,t} \sim N(\log\lambda_{V,t-1}, 1/\kappa)$.

Importantly, only the visual reliability, $\lambda_{V,t}$, is directly dependent on the previous trial $(P(\lambda_{V,k},\lambda_{V,k-1}) = P(\lambda_{V,k}|\lambda_{V,k-1})P(\lambda_{V,k-1}) \neq P(\lambda_{V,k})P(\lambda_{V,k-1}))$. Because of the Markov property (i.e. $\lambda_{V,t}$ depends only on $\lambda_{V,t-1}$), the joint distribution for time $t$ can be written as

$$P(\lambda_{V,t}, \lambda_{V,t-1}, A_t, U_t, V_{1:n,t}, S_t)$$
$$= P(A_t|S_t)P(U_t|S_t,\lambda_{V,t})P(V_{1:n}|U_t,\lambda_{V,t})P(\lambda_{V,t}|\lambda_{V,t-1})P(\lambda_{V,t-1}|V_{1:n,t-1},A_{t-1})P(S_t). \tag{9}$$

Hence, the joint posterior probability over location and visual reliability given a stream of auditory and visual inputs can be rewritten as:

$$P(S_t, U_t, \lambda_{V,t}|A_{1:t}, V_{1:n,1:t}) =$$
$$P(S_t)P(A_t|S_t)P(U_t|S_t,\lambda_{V,t})P(V_{1:n}|U_t,\lambda_{V,t})\int P(\lambda_{V,t}|\lambda_{V,t-1})P(\lambda_{V,t-1}|V_{1:n,t-1},A_{t-1})d\lambda_{V,t-1}/Z. \tag{10}$$

As this equation cannot be solved analytically, we obtain an approximate solution by factorizing the posterior in terms of the unknown variables $(S_t, U_t, \lambda_{V,t})$ according to the method of variational Bayes (**Bishop, 2006**). In this approximate method (for details see Appendix 2), the posterior is factorized into three terms, each a normal distribution:

$$P(S_t, U_t, \lambda_{V,t}|A_t, V_{1:n,t}) \approx q(S_t, U_t, \lambda_{V,t}) = q(S_t)^\star q(U_t)^\star q(\lambda_{V,t_t}).$$

In order to estimate the set of parameters (mean and variance) of $q(S_t)$, $q(U_t)$ and $q(\lambda_{V,t}t)$, the Free Energy is minimized iteratively (and thereby the Kullback–Leibler divergence between the true

and approximate distribution), until a convergence criterion is reached (here, the change in each fitted parameter is less than 0.0001 between iterations).

This is done separately for the common cause model component (C = 1) and the independent cause model component (C = 2). The auditory location, for the common cause model is based on the approximation over the posterior location of $\hat{S}_{A,C=1,t}$ from, $q_1(S_t) = N(\hat{S}_{A,C=1,t}, \sigma_{1,t})$. The auditory location for the independent cause model is simply computed as $\hat{S}_{A,C=2,t} = A_t / (1 + \sigma_A^2 / \sigma_0^2)$, because it is independent of the visual signal.

The marginal model evidence is estimated based on the minimized Free Energy for each mode component, $P(A_t, V_{1:n,t} | C = 1)$, respectively $P(A_t, V_{1:n,t} | C = 2)$ to form the posterior probability $P(C = 1 | A_t, V_{1:n,t})$, as described above in *Equation 4*. These values can then be used to compute the predicted responses for a particular participant according to *Equation 3*.

## Model 3: Exponential learner

Finally, the observer may approximate the full Bayesian inference of the Bayesian learner by a more simple heuristic strategy of exponential discounting. In the exponential discounting model, the observer learns the visual reliability by exponentially discounting past visual reliability estimates:

$$\hat{\lambda}_{V,t-1} = 1/\sigma_{Vt}^2 (1 - \gamma) + \hat{\lambda}_{V,t-1} \gamma \tag{11}$$

where $\sigma_{Vt}^2 = 1/(n-1) \sum_{i=1}^{n} (V_{i,t} - \bar{V}_{i,t})^2$ is the sample variance and $\bar{V}_t = 1/n \sum_{i=1}^{n} V_{i,t}$ is the sample mean.

Similar to the optimal Bayesian learner (above), this observer model uses the past to compute the current reliability, but it does so based on a fixed learning rate 1 - γ. Computation is otherwise performed in accordance with models 1 and 2, *Equations 3-4 and 6-7*.

## Assumptions of the computational models: motivation and caveats

Computational models inherently make simplifying assumptions about the generation of the sensory inputs and observers' inference.

First, we modeled that visual signals (i.e. the cloud's mean) were sampled from a Gaussian, while they were sampled from a uniform discrete distribution (i.e. [−10°, −5°, 0°, 5°, 10°]) in the experiment. Gaussian assumptions about the stimuli locations have nearly exclusively been made in the recent series of studies focusing on Bayesian Causal Inference in multisensory perception (*Körding et al., 2007*; *Rohe and Noppeney, 2015b*; *Rohe and Noppeney, 2015a*). Because visual signals have been sampled from a wide range of visual angle (i.e. 20°) and are corrupted by physical (i.e. cloud of dots) and internal neural noise, we used the simplifying assumption of a Gaussian spatial prior consistent with previous research.

Second, we assumed that the auditory signal location is sampled from a Gaussian, while the experiments presented sounds ±5° from the visual location. These Gaussian assumptions about sound location can be justified by the fact that observers are known to be limited in their sound localization ability, particularly when generic HRTFs were used to generate spatial sounds. Moreover, because sounds are presented together with visual signals, it is even harder for observers to obtain an accurate estimate of the sound's location.

Third, in the experiment we generated the cloud of dots directly from a Gaussian distribution centred on $S_t$. By contrast, in the model we introduced a hidden variable $U_t$ that is sampled from a Gaussian centred on $S_t$. The visual cloud of dots is then centred on this hidden variable $U_t$. We introduced this additional hidden variable $U_t$ to account for observers' additional causal uncertainty in natural environments, in which even signals from a common source may not fully coincide in space. Critically, the dispersion of the cloud of dots is set to be equal to the STD of the distribution from which $U_t$ is sampled, so that the cloud's STD informs observers about the variance of the hidden variable $U_t$.

## Inference by the experimenter

From the observer's viewpoint, this completes the inference process. However, from the experimenter's viewpoint, the internal variable for the auditory stimulus, $A_t$, is unknown and not directly under the experimenter's control. To integrate out this unknown variable, we generated 1000 samples of

the internal auditory value for each trial from the generative process $A_t \sim N(S_{A,t,true}, \sigma_A^2)$, where $S_{A,t,true}$ was the true location the auditory stimulus came from. For each value of $A_t$, we obtained a single estimate $\hat{S}_{A,t}$ (as described above). To link these estimates with observers' button response data, we assumed that subjects push the button associated with the position closest to $\hat{S}_{A,t}$. In this way, we obtained a histogram of responses for each subject and trial which provide the likelihood of the model parameters given a subject's responses: $P(resp_t|\kappa, \sigma_A, P_{common}, S_{A,t,true}, S_{V,t,true})$.

## Model estimation and comparison

Parameters for each model (for all models: $\sigma_A$, $P_{common}$ = P(C = 1), $\sigma_0$, Bayesian learner: $\kappa$, exponential learner: $\gamma$) were fit for each individual subject by sampling using a symmetric proposal Metropolis-Hasting (MH) algorithm (with $A_t$ integrated out via sampling, see above). The MH algorithm iteratively draws samples $set_n$ from a probability distribution through a variant of rejection sampling: if the likelihood of the parameter set is larger than the previous set, the new set is accepted, otherwise it is accepted with probability $L(model|set_n)/L(model|set_{n-1})$, where $L(resp|set_n) =$ $\prod_t P(resp_t|\kappa, \sigma_A, P_{common}, S_{A,t,true}, S_{V,t,true})$ (for Bayesian learner). We sampled 4000 steps from four sampling chains with thinning (only using every fourth sampling to avoid correlations in samples), giving a total of 4000 samples per subject data sets. Convergence was assessed through scale reduction (using criterion R < 1.1 [*Gelman et al., 2013*]). Using sampling does not just provide a single parameter estimate for a data set (as when fitting maximum likelihood), but can instead be used to assess the uncertainty in estimation for the data set. The model code was implemented in Matlab (Mathworks, MA) and ran on two dual Xeon workstations. Each sample step, per subject data set, took 30 s on a single core (~42 hr per sampling chain).

Quantitative Bayesian model comparison of the three candidate models was based on the Watanabe-Akaike Information Criterion (WAIC) as an approximation to the out of sample expectation (*Gelman et al., 2013*). At the fixed-effects level, Bayesian model comparison was performed by summing the WAIC over all participants within each experiment. For a random-effects analysis, we transformed the WAIC into log-likelihoods by dividing them by minus 2. We then computed the protected exceedance probability that one model is better than the other model beyond chance using hierarchical Bayesian model selection (*Penny et al., 2010*; *Rigoux et al., 2014*).

To qualitatively compare the localization responses given by the participants and the responses predicted by the instantaneous, Bayesian and exponential learner, we computed the auditory weight $w_A$ from the predicted responses of the three models exactly as in the analysis for the behavioral data. For illustration, we show and compare the model's $w_A$ from the 1st and the flipped 2nd half of the periods for each of the four experiments (*Figure 3*, *Figure 4*, *Figure 5B/C* and *Figure 5—figure supplement 1*).

## Parameter recovery

To test the validity of the models, we performed parameter recovery and were able to recover the generating values with a bias of all parameters smaller than 10% (for full details of bias and variance across parameters, see Appendix 1 and *Supplementary file 1-Table 7*).

## Simulated localization responses

To further compare the Bayesian and exponential learner and assess whether they can be discriminated experimentally, we simulated the choices of 12 subjects for the continuous sinusoidal and sinusoidal jump sequence using the Bayesian learner model (parameters: $\sigma_A = 6°$, $\kappa = 15$, $P_{common} = 0.7$ and $\sigma_0 = 12°$). To increase observers' uncertainty about their visual reliability estimates, we reduced the number of dots in the visual clouds from 20 to 5 dots where we ensured that the mean and variance of the five dots corresponded to the experimentally defined visual mean and variance. We then fitted the Bayesian learner and exponential learner models to each simulated data set (using the BADS toolbox for likelihood maximization [*Acerbi and Ma, 2017*]). The fitted parameters for the Bayesian model, $set_{Bayes}$ were very close to the parameters used to generate observers' simulated responses (sinusoidal sequence, fitted parameters: $\sigma_A = 6.11°$, $\kappa = 17.5$, $P_{common} = 0.72$ and $\sigma_0 = 12.4°$; sinusoidal jump sequence, fitted parameters: $\sigma_A = 6.08°$, $\kappa = 17.3$, $P_{common} = 0.71$ and $\sigma_0 = 12.2°$) – thereby providing a simple version of parameter recovery. The parameters of the

exponential model, $set_{Exp}$ (fitted to observers' responses generated from the Bayesian model) were very similar to those of the Bayesian learner (sinusoidal sequence: $\sigma_A = 5.99°$, $\gamma = 0.70$, $P_{common} = 0.61$ and $\sigma_0 = 12.0°$, sinusoidal jump sequence: $\sigma_A = 6.06°$, $\gamma = 0.70$, $P_{common} = 0.65$ and $\sigma_0 = 12.0°$). Moreover, the fits to the simulated observers' responses were very close for the two models (*Figure 6*), with mean log likelihood difference $(log(L(resp|set_{Bayes})) - log(L(resp|set_{Exp}))) = 1.82$ for the sinusoidal and 2.74 for the sinusoidal jump sequence (implying a slightly better fit for the Bayesian learner). *Figure 6C and D* show the timecourses of observers' visual uncertainty (STD) as estimated by the Bayesian and exponential learners.

## Acknowledgements

This study was funded by the ERC (ERC-multsens, 309349), the Max Planck Society and the Deutsche Forschungsgemeinschaft (DFG; grant number RO 5587/1–1). We thank Peter Dayan for his valuable contributions and very helpful comments on a previous version of the manuscript.

## Additional information

### Funding

| Funder | Grant reference number | Author |
| --- | --- | --- |
| European Research Council | ERC-multsens,309349 | Uta Noppeney |
| Max Planck Society | | Tim Rohe<br>Uta Noppeney |
| Deutsche Forschungsge-meinschaft | DFG RO 5587/1-1 | Tim Rohe |

The funders had no role in study design, data collection and interpretation, or the decision to submit the work for publication.

### Author contributions

Ulrik Beierholm, Conceptualization, Resources, Software, Formal analysis, Investigation, Visualization, Methodology, Writing - review and editing; Tim Rohe, Conceptualization, Resources, Data curation, Formal analysis, Funding acquisition, Investigation, Visualization, Methodology, Writing - original draft, Project administration, Writing - review and editing; Ambra Ferrari, Data curation, Project administration; Oliver Stegle, Conceptualization, Methodology; Uta Noppeney, Conceptualization, Formal analysis, Supervision, Funding acquisition, Investigation, Methodology, Writing - review and editing

### Author ORCIDs

Ulrik Beierholm https://orcid.org/0000-0002-7296-7996
Tim Rohe https://orcid.org/0000-0001-9713-3712
Ambra Ferrari http://orcid.org/0000-0003-1946-3884

### Ethics

Human subjects: All volunteers participated in the study after giving written informed consent. The study was approved by the human research review committee of the University of Tuebingen (approval number 432 2007 BO1) and the research review committee of the University of Birmingham (approval number ERN_15-1458AP1).

### Decision letter and Author response

Decision letter https://doi.org/10.7554/eLife.54172.sa1
Author response https://doi.org/10.7554/eLife.54172.sa2

# Additional files

## Supplementary files

- Supplementary file 1. Seven tables showing results of additional analyses.
- Transparent reporting form

## Data availability

The human behavioral raw data and computational model predictions as well as the code for computational modelling and analyses scripts are available in an OSF repository: https://osf.io/gt4jb/.

The following dataset was generated:

| Author(s) | Year | Dataset title | Dataset URL | Database and Identifier |
|---|---|---|---|---|
| Beierholm U, Rohe T, Noppeney U | 2020 | Using the past to estimate sensory uncertainty | https://osf.io/gt4jb/ | Open Science Framework, 10.17605/OSF.IO/GT4JB |

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

# Appendix 1

## Additional methods and results

Influence of the visual location of the previous trial on observers' sound localization responses

We have performed a control regression analysis to assess the influence of the visual location of the previous trial on observers' sound localization response. This is important because 200 ms prior to trial onset and sound presentation, observers were presented with a visual cloud whose mean was the same as for the previous trial and the cloud's standard deviation (STD) varied according to a continuous or discontinuous sequence (see main paper). To quantify the influence of the previous visual location, we expanded our regression model that we used in the main paper by another regressor modeling the visual cloud's location on the previous trial. For instance, for bin = 1, we computed:

$$R_{A,\text{trial,bin}=1} = L_{A,\text{trial,bin}=1}\, \ss_{A,\text{bin}=1} + L_{V,\text{trial,bin}=1}\, \ss_{V,\text{bin}=1}$$
$$+ L_{V,\text{trial}-1,\text{bin}=1}\, \ss_{V\text{previous,bin}=1} + \ss_{\text{const,bin}=1} + e_{\text{trial,bin}=1}$$

with $R_{A,\text{trial, bin}=1}$ = localization response for current trial that is assigned to bin 1; $L_{A,\text{trial,bin}=1}$ or $L_{V,\text{trial,bin}=1}$= 'true' auditory or visual location for current trial that is assigned to bin 1; $L_{V,\text{trial-1,bin}=1}$ 'true' visual location for corresponding previous trial (for explanatory purposes, we assign here the bin of the current trial; the previous trial actually falls into a different bin); $\ss_{A,\text{bin}=1}$ or $\ss_{V,\text{bin}=1}$ = quantified the influence of the auditory and visual location of the *current* trial on the perceived sound location of the current trial for bin 1; $\ss_{V\text{previous,bin}=1}$ quantified the influence of the visual location of the *previous* trial on the perceived sound location of the current trial for bin 1. $\ss_{\text{const,bin}=1}$ = constant term; $e_{\text{trial,bin}=1}$ = error term. For each bin, we thus obtained another visual weight estimate $\ss_{V\text{previous,bin}}$ for the previous location.

First, we averaged $\ss_{V\text{previous,bin}}$ across bins and entered these participant-specific bin-averaged $\ss_{V\text{previous}}$ into two-sided one-sample t-tests at the between-subject random effects level. Results: As shown in *Supplementary file 1-Table 2A*, this analysis demonstrated that the visual location of the previous trial significantly influenced observers' perceived sound location on the current trial in the Sinusoidal, RW1 and (marginally) the Sinusoidal jump sequence.

Second, we computed the correlation of $\ss_{V\text{previous,bin}}$ with the visual noise in the *current* trials averaged in a given bin (r($\ss_{V\text{previous,bin}}$, $\sigma_{V\text{current,bin}}$)) or with the visual noise (i.e. visual cloud's STD) in the *previous* trial averaged within a given bin (r($\ss_{V\text{previous,bin}}$, $\sigma_{V\text{previous,bin}}$)). The correlations were computed over bins within each participant. We entered the participant-specific Fisher z-transformed correlation coefficients into two-sided one-sample t-tests at the between-subject random-effects level (see *Supplementary file 1-Table 2*, section B and C). Results: These analyses demonstrated that the influence of the visual location on the previous trial was not correlated with the visual cloud's STD on the current or previous trial (apart from one significant p-value for the sinusoidal jump sequence, but after Bonferroni correction for the eight additional statistical comparisons this p-value is no longer statistically significant either). This analysis already suggests that the previous visual location is unlikely responsible for the effects of the previous STD on observers' perceived sound location.

Third and most importantly, this regression model provides us with weights for the auditory ($\ss_{A,\text{bin}}$) and visual ($\ss_{V,\text{bin}}$) locations in the current trial from a regression model that regressed out the influence of the previous visual location. We used those auditory ($\ss_{A,\text{bin}}$) and visual ($\ss_{V,\text{bin}}$) weights as in the main paper to compute bin-specific $w_{A,\text{bin}} = \ss_{A,\text{bin}} / (\ss_{A,\text{bin}} + \ss_{V,\text{bin}})$. Following exactly the same procedures as in the main paper, we then assessed in a repeated-measures ANOVA whether these $w_{A,\text{bin}}$ differed between first and second half (see *Supplementary file 1-Table 3*). Moreover, we repeated a second regression model analysis to assess whether $w_{A,\text{bin}}$ was predicted not only by the visual cloud's STD of the current, but also of the previous bin using the following regression model (i.e. *Equation 2* in the main text): $w_{A,\text{bin}} = \sigma_{V,\text{bin}} * \ss_{\sigma V} + (\sigma_{V,\text{bin}} - \sigma_{V,\text{bin-1}}) * \ss_{\Delta\sigma V} + \ss_{\text{const}} + e_{\text{bin}}$ with $w_{A,\text{bin}}$ = relative auditory weight in bin b; $\sigma_{V,\text{bin}}$ = mean visual STD in current bin b or previous bin b-1; $\ss_{\text{const}}$ = constant term; $e_{\text{bin}}$ = error term. To allow for generalization to the population level, the parameter estimates ($\ss_{\sigma V}$, $\ss_{\Delta\sigma V}$) for each subject were entered into two-sided one-sample t-tests at the between-subject random-effects level (see *Supplementary file 1-Table 4*). Results: These control analyses (see *Figure 2—figure supplement 1*, *Supplementary file 1-Table 4*) replicate our

initial analyses reported in the main manuscript. Collectively, they provide further evidence that the effect of previous visual location on observers' perceived sound location cannot explain the effect of prior visual reliability that are the key focus of our paper.

## Nested model comparison to assess the effect of past visual noise on observers' auditory weights

To assess the effect of past visual noise on auditory weights (*Equation 2* in the main text), we also formally compared two nested linear mixed-effects models to predicts observers' relative auditory weights $w_{A,bin}$ separately for the Sin, RW1, and RW2 sequences. The reduced model included only the STD of the current bin as fixed effects. The full model included both the STD of the current bin and the difference in STD between the current and the previous bin as fixed effects. Both the reduced and the full model included participants as random effects. After fitting the two models using maximum likelihood estimation, we compared them using loglikelihood ratio tests and the Bayesian Inference Criterion as an approximation to the model evidence (see *Supplementary file 1-Table 5*). The model comparison demonstrated that the full model including the difference in STD provided a better explanation of observers relative auditory weights $w_{A,bin}$ across all four sequences. This corroborates that observers estimate sensory uncertainty by combining information from past and current sensory inputs.

## Characterization of observer's behavior and the model predictions before and after the jumps in visual reliability

One critical question in the discontinuous sinusoidal jump sequence is whether observers continue combining past with current inputs to adapt their visual uncertainty estimates. One may argue that observers detect the discontinuity, 'reset' the estimation of visual uncertainty after the jump and therefore do not integrate information from before the jump. In that case Bayesian learning models would not be ideal for modeling observers' behavior in the jump sequence and better accommodated by Bayesian change-point detection model (*Adams and Mackay, 2007*; *Heilbron and Meyniel, 2019*).

First, we inserted the jumps selectively in the period sections in which the visual variance was greater, so that changes in visual variance were more difficult to detect. This experimental choice minimized the chances that observers 'reset' their estimation of visual uncertainty.

Second, we assessed observers' estimation strategy at a greater temporal resolution before and after the jumps. To estimate the relative auditory weight $w_A$ at a greater resolution, we applied separate regression models to individual sampling points of the visual cloud of dots presented every 200 ms (i.e. no binning). Thus, $w_A$ was computed at 5 Hz resolution before and after the jumps (i.e. at time points $[-1.9:0.2:1.9]$ s; *Figure 5—figure supplement 2*). Because the number of trials was very low on individual sampling points, we pooled the trials across the three up- and down-jumps before computing the regression models. Nevertheless, the small number of trials on individual sampling points (range 7–28 trials across participants and bins) rendered the estimation of the relative auditory weights very unreliable. Thus, individual $w_A$ values that were smaller or larger than three times the scaled median absolute deviation were excluded from the analyses in *Figure 5—figure supplement 2* and *Supplementary file 1-Table 6*. To assess statistically whether participants' adjusted $w_A$ after the jump, we computed a paired t test on $w_A$ specifically from the time point before versus after the jump (i.e. $-0.1$ vs. $0.1$ s; *Supplementary file 1-Table 6*).

Third, we assessed the model fit of our learning models before and after the jumps. If the jumps violate the assumptions of the learning models, we would expect that observers' behavior deviates from the model's predictions more strongly after the jump. We computed the root mean squared error of the models' $w_A$ (i.e. $(w_{A,behavior} - w_{A,model})^2$) before and after the change point (*Figure 5—figure supplement 2B*) and entered those into a paired t test (i.e. $-0.1$ vs. $0.1$ s; *Supplementary file 1-Table 6*).

Results: Our data showed that participants and models rapidly and significantly adjusted their weights after the jumps. Critically, the model fits did not significantly differ for the time points just before or after the jumps in visual variance (i.e. if anything, they significantly decreased after the jump; *Figure 5—figure supplement 2B* and *Supplementary file 1-Table 6*). Collectively, these

control analyses suggest that our Bayesian and exponential learning models adequately modeled observers' visual uncertainty adaptation both before and after change points (*Norton et al., 2019*).

## Parameter recovery

To test the validity of the Bayesian, exponential, and instantaneous models, we performed parameter recovery by assessing the bias and variability of the parameters fitted to simulated data sets with respect to the true parameters used to generate the data.

For each model, we selected four different parameter sets (within a realistic range of values for parameters $\sigma_A = [6:12]°$, $P_{common} = [0.7:0.9]$, $\sigma_0 = [6:20]°$, $\kappa = [5:20]$, $\gamma = [0.3:0.7]$) and generated data sets of simulated observers for the RW2 sequence. We repeated this process six times (with different initial random seeds), creating a total of 24 simulated data sets for each model. We then fitted the Bayesian, exponential, and instantaneous learner models to each simulated data set (using exactly the same fitting procedures as for observers' data in the experiments) resulting in 24 sets of best fitting parameters for each model.

In order to assess how well the fitting procedure recovers the generating parameters, we compared the fitted parameters to the 'true' parameters used to generate the data. Specifically, we assessed the parameter recovery in terms of bias and variability of the fitted parameters as follows: The recovered parameters' bias was computed as the signed deviation from the true generating value in percentage. As an example, if a data set was generated with a model parameter of 5, but the fitted (i.e. recovered) parameter was 4, we would compute a $-20\%$ deviation. As a measure of the variability for the recovered parameters, we calculated the absolute (i.e. unsigned) deviation from the true generating values in percentage. As an example, a fitted value of 4, relative to a generating value of 5 would be a 20% absolute deviation. We report the median (and first and third quartile) across simulated data sets as a robust measure for this bias and variability.

## Appendix 2

This document describes a Variational Bayes approximation to inference on a generative model that allows for two possible ways that stimuli data was generated (thus allowing subjects to perform causal inference).

Section (1) describes the full generative model for both a single and two sources, section (2) explains how an optimal observer can perform inference within either sub-model, through a variational Bayes approximation to the posteriors and section (3) shows how to calculate the model likelihood for either sub-model, as necessary for combining the two sub-models.

Section (4) finally describes how the results for each causal model are combined into a single posterior.

### 1 Generative model

The model presented here is an extension of the Causal Inference model of *Körding et al., 2007*, with the reliability of the visual signal assumed to be changing smoothly over trials according to a random walk (RW). In the case where the visual reliability is constant the model approximates the original Causal Inference model.

In this model (figure *Appendix 2—figure 1*), at each stimulus presentation, $t$, subjects assume that the visual dots at positions $V_{i,t}$ and auditory stimulus at $A_t$, are generated through either of two causal models ($C_t = 1$ or $C_t = 2$) with fixed prior probabilities:

$$C_t \sim Bernoulli(pcommon) \tag{1}$$

If $C_t = 1$, (single source, $S_t$, leading to forced fusion)

$$S_t \sim \mathcal{N}(S_t; \mu_0, \sigma_0^2) \tag{2}$$
$$A_t \sim \mathcal{N}(A_t; S_t, \sigma_A^2) \tag{3}$$
$$U_t \sim \mathcal{N}(U_t; S_t, 1/\lambda_{V,t}) \tag{4}$$
$$V_{i,t} \sim \mathcal{N}(V_{i,t}; U_t, 1/\lambda_{V,t}) \tag{5}$$
$$\log \lambda_{V,t} \sim \mathcal{N}(\log(\lambda_{V,t}); \log(\lambda_{V,t-1}), k) \tag{6}$$

If $C_t = 2$ (independent sources, $S_{A,t}$ and $S_{V,t}$ )

$$S_{A,t} \sim \mathcal{N}(S_{A,t}; \mu_0, \sigma_0^2) \tag{7}$$
$$A_t \sim \mathcal{N}(A_t; S_{A,t}, \sigma_A^2) \tag{8}$$
$$S_{V,t} \sim \mathcal{N}(S_{V,t}; \mu_0, \sigma_0^2) \tag{9}$$
$$U_t \sim \mathcal{N}(U_t; S_{V,t}, 1/\lambda_{V,t}) \tag{10}$$
$$V_{i,t} \sim \mathcal{N}(V_{i,t}; U_t, 1/\lambda_{V,t}) \tag{11}$$
$$\log \lambda_{V,t} \sim \mathcal{N}(\log(\lambda_{V,t}); \log(\lambda_{V,t-1}), k) \tag{12}$$

The intermediate variable $U_t$ means that the mean of the visual dots is not located at the true source ($S_t$ or $S_{V,t}$), but normally distributed around it. Note that for $C_t = 1$ we can explicitly write $S_{A,t} = S_{V,t} = S_t$.

For simplicity, we assume that $\mu_0 = 0$, that is, the prior mean is located at the horizontal center. The auditory STD $\sigma_A$, the prior probability of a single cause, $P_{common}$ and the prior STD, $\sigma_0$ , are fixed individually for each subject (see main text for fitting procedure).

In the following, we will simplify the notation by referring to $P(*|C_t = 1)$ by $P_1(*)$ and $P(*|C_t = 2)$ by $P_2(*)$.

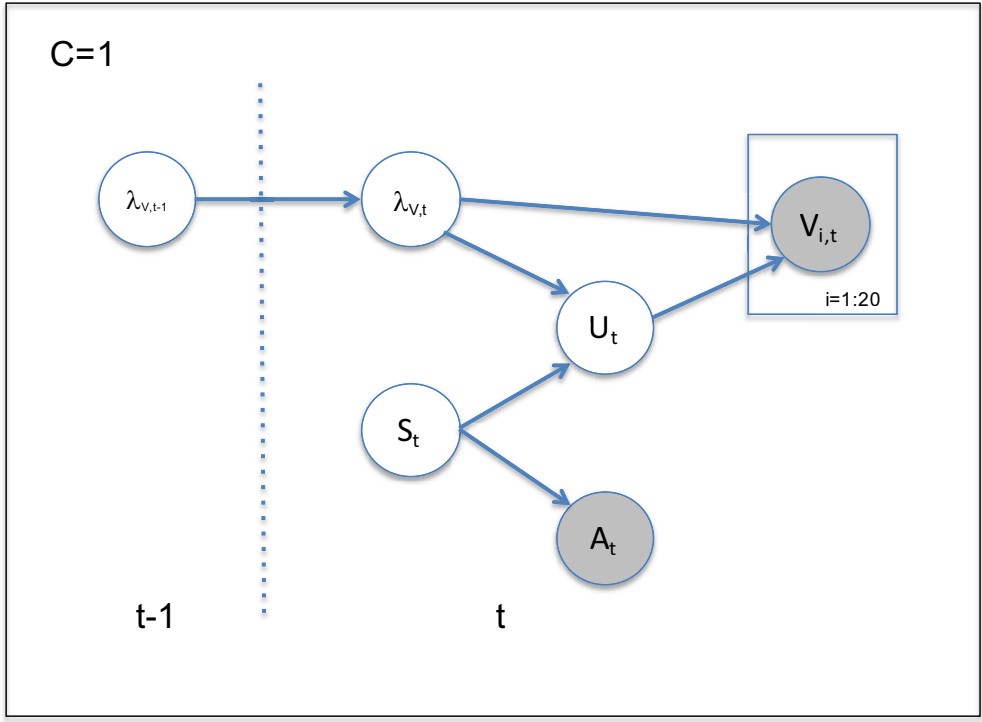

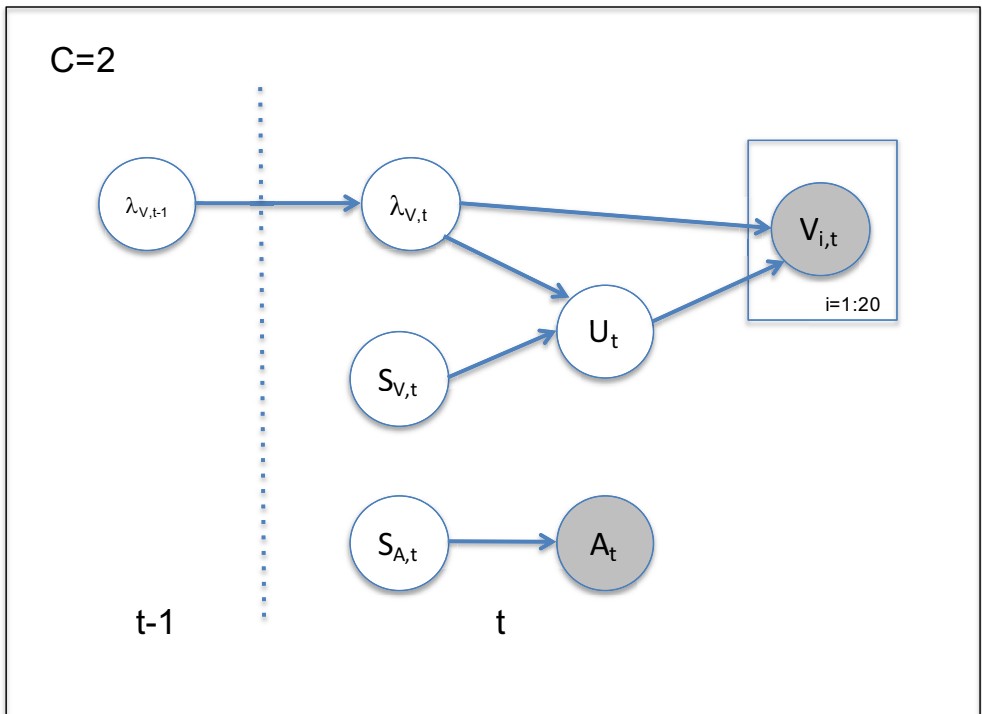

**Appendix 2—figure 1.** Generative model, for one (C = 1) or two sources (C = 2).

## 2 Posterior

The full posterior over the latent variables in the model up until time $t$ is

$$P(S_{V,1:t}, S_{A,1:t}, \lambda_{1:t}, U_{1:t}, C_{1:t}|A_{1:t}, V_{1:N,1:t}) =$$
$$P(S_{V,1}, S_{A,1}, \lambda_{V,1}, U_1, C_1|A_1, V_{1:N,1}) \prod_{j=1}^{t} P(S_{V,j}, S_{V,j}, \lambda_{V,j}, U_j, C_j|A_{1:j}, V_{1:N,1:j}, \lambda_{V,j-1}) \tag{13}$$

Recursively we can write

$$P(S_{V,t}, S_{A,t}, \lambda_{V,t}, U_t, C_t|A_{1:t}, V_{1:N,1:t}) = \int P(S_{V,t}, S_{A,t}, \lambda_{V,t}, U_t, C_t|A_{1:t}, V_{1:N,1:t}, \lambda_{V,t-1})$$
$$P(S_{V,t-1}, S_{A,t-1}, U_{t-1}, C_t, \lambda_{V,1:t-1}|A_{1:t-1}, V_{1:N,1:t-1}) dS_{V,t-1} dS_{A,t} dU_{t-1} d\lambda_{V,1:t-1} \tag{14}$$

$$P(S_{V,t}, S_{A,t}, \lambda_{V,t}, U_t, C_t|A_{1:t}, V_{1:N,1:t}) \propto P(C_t)P(S_{A,t}, S_{V,t})P(A_t|S_{A,t})$$
$$P(V_{1:N,t}|U_t, \lambda_{V,t})P(U_t|S_{V,t}, \lambda_{V,t}) \tag{15}$$
$$\int P(\lambda_{V,t}|\lambda_{V,t-1})P(\lambda_{V,1:t-1}|A_{1:t-1}, V_{1:N,1:t-1})d\lambda_{V,1:t-1}/Z$$

where $P(S_{A,t}, S_{V,t})$ obviously depends on $C_t$ through the generative model (*Appendix 2—figure 1*). If we marginalize over the latent $C_t$:

$$P(S_{V,t}, S_{A,t}, \lambda_{V,t}, U_t|A_{1:t}, V_{1:N,1:t}) =$$
$$P(S_{V,t}, S_{A,t}, \lambda_{V,t}, U_t|A_{1:t}, V_{1:N,1:t}, C_t = 1)P(C_t = 1|A_{1:t}, V_{1:N,1:t})$$
$$+P(S_{V,t}, S_{A,t}, \lambda_{V,t}, U_t|A_{1:t}, V_{1:N,1:t}, C_t = 2)P(C_t = 2|A_{1:t}, V_{1:N,1:t}) \tag{16}$$

At this point it should be clear that the posterior is a mixture of the forced fusion and independent solutions, with the mixture determined by the posterior probability of either model generating the data:

$$P(C_t = 1|A_{1:t}, V_{1:N,1:t}) = \frac{P(A_t, V_{1:N,t}|C_t = 1)P(C_t = 1)}{P(A_t, V_{1:N,t}|C_t = 1)P(C_t = 1) + P(A_t, V_{1:N,t}|C_t = 2)P(C_t = 2)} \tag{17}$$

To evaluate this, we need to calculate the marginal model evidence, $P(A_t, V_{1:N,t}|C_t)$, for either model, see the later section.

## 2.1 Posterior for C = 1

The full posterior over the latent variables in the single source sub-model is

$$P_1(S_{1:t}, \lambda_{1:t}, U_{1:t}|A_{1:t}, V_{1:N,1:t}) = P(S_1, \lambda_{V,1}, U_1|A_1, V_{1:N,1}) \prod_{i=1}^{t} P(S_i, \lambda_{V,i}, U_i|A_i, V_i, \lambda_{V,i-1}) \tag{18}$$

Recursively we can write

$$P_1(S_t, \lambda_{V,t}, U_t|A_{1:t}, V_{1:N,1:t}) = \int P_1(S_t, \lambda_{V,t}, U_t|A_{1:t}, V_{1:N,1:t}, \lambda_{V,t-1})$$
$$P_1(S_{t-1}, \lambda_{V,1:t-1}, U_{t-1}|A_{1:t-1}, V_{1:N,1:t-1}) dS_{t-1} dU_{t-1} d\lambda_{V,1:t-1} \tag{19}$$

$$P_1(S_t, \lambda_{V,t}, U_t|A_{1:t}, V_{1:N,1:t}) = \int P_1(S_t, \lambda_{V,t}, U_t|A_{1:t}, V_{1:N,1:t}, \lambda_{V,t-1})$$
$$P(\lambda_{V,1:t-1}|A_{1:t}, V_{1:N,1:t-1})d\lambda_{V,1:t-1} \tag{20}$$

$$P_1(S_t, \lambda_{V,t}, U_t|A_{1:t}, V_{1:N,1:t})(S_t)P(A_t|S_t)P(U_t|S_t, \lambda_{V,t})P(V_{1:N,t}|U_t, \lambda_{V,t})$$
$$\int P(\lambda_{V,t}|\lambda_{V,t-1})P(\lambda_{V,1:t-1}|A_{1:t-1}, V_{1:N,1:t-1})d\lambda_{V,1:t-1}/Z \tag{21}$$

As we will see later it is convenient to use a change of parameters

$$\theta_t = \log(\lambda_{V,t}) \tag{22}$$

allowing us to rewrite

$$P_1(S_t, \theta_t, U_t|A_{1:t}, V_{1:N,1:t}) \propto P(S_t)P(A_t|S_t)P(U_t|S_t, \theta_t)P(V_{1:N,t}|U_t, \theta_t)$$
$$\int P(\theta_t|\theta_{V,t-1})P(\theta_{V,1:t-1}|A_{1:t-1}, V_{1:N,1:t-1})d\theta_{V,1:t-1}/Z \tag{23}$$

where

$$P(U_t|S_t, \theta_t) = \mathcal{N}(U_t; S_t, 1/\exp(\theta_t)) \tag{24}$$

$$P(V_{1:N,t}|U_t, \theta_t) = \prod_n \mathcal{N}(U_t; V_{n,t}, 1/\exp(\theta_t)) \tag{25}$$

$$P(\theta_t|\theta_{V,t-1}) = \mathcal{N}(\theta_t; \theta_{V,t-1}, 1/\kappa) \tag{26}$$

We will assume that

$$P(\theta_{V,1:t-1}|A_{1:t-1}, V_{1:N,1:t-1}) \tag{27}$$

can be approximated by a Normal distribution (see below), thus allowing us to write

$$\int P(\theta_t|\theta_{V,t-1})P(\theta_{V,1:t-1}|A_{1:t-1}, V_{1:N,1:t-1})d\theta_{V,1:t-1} = \mathcal{N}(\theta_t|\theta_{V,t-1}, 1/\kappa'_t) \tag{28}$$

where $1/\kappa'_t = 1/\kappa + 1/\tau_{\theta,t-1}$ (due to properties of convolution of two Normal distributions). The log-posterior (to be used for a variational approximation) is now

$$
\begin{aligned}
\log P_1(S_t, \lambda_{V,t}, U_t|A_{1:t}, V_{1:N,1:t}) \propto\ & -\lambda_S S_t^2/2 \\
& -\lambda_A(A_t - S_t)^2/2 \\
& -\lambda_{V,t}(U_t - S_t)^2/2 + \log\lambda_{V,t}/2 \\
& -\lambda_{V,t}\sum_i^N(U_t - V_{i,t})^2/2 + (N/2)\log\lambda_{V,t} \\
& -\kappa'(\theta_t - \theta_{V,t-1})^2/2
\end{aligned}
\tag{29}
$$

## 2.2 Variational Bayes approximation for C = 1

We will now approximate the log-posterior with variational Bayes by factorization:

$$P_1(S_t, \theta_t, U_t|A_t, V_{1:N,t}) \approx q_1(S_t, U_t, \theta_t) = q_1(S_t) * q_1(U_t) * q_1(\theta_t) \tag{30}$$

### 2.2.1 $q_1(S_t)$

For $q_1(S_t)$

$$\log q_1(S_t) \propto -\lambda_S S_t^2/2 - \lambda_A(A_t - S_t)^2/2 - E_\theta(\exp(\theta_t))E_U((U_t - S_t)^2)/2 \tag{31}$$

where $E_Y(X)$ signifies the expectation of X over the distribution of Y: $E_Y(X) = \int P(Y)XdY$.

Using $\quad E_U((U_t - S_t)^2) = E_U(U_t^2 + S_t^2 - 2U_tS_t) = E_U(U_t^2) + S_t^2 - 2S_tE_U(U_t) \quad =$ $(S_t^2 - S_tE_U(U_t) * 2 + E_U(U_t)^2) - E_U(U_t)^2 + E_U(U_t^2) = (S_t - E_U(U_t))^2 - E_U(U_t)^2 + E_U(U_t^2)$, where the last two terms do not depend on $S_t$ (and thus can be discarded) we can rewrite the last term:

$$\log q_1(S_t) \propto -\lambda_S S_t^2/2 - \lambda_A(A_t - S_t)^2/2 - E_\theta(\exp(\theta_t))(S_t - E_U(U_t))^2/2 \tag{32}$$

### 2.2.2 $q_1(U_t)$

For $q_1(U_t)$

$$\log q_1(U_t) \propto -E_\theta(\exp(\theta_t))/2\sum_i(U_t - V_{i,t})^2 - E_\theta(\exp(\theta_t))E_S((U_t - S_t)^2)/2 \tag{33}$$

Here, we use the same trick

$$\log q_1(U_t) \propto -E_{\theta_t}(\exp(\theta_t))/2\sum_i(U_t - V_{i,t})^2 - E_\theta(\exp(\theta_t))\left(U_t - E_S(S_t)\right)^2/2 \tag{34}$$

### 2.2.3 $q_1(\theta)$

For $q_1(\theta)$

$$
\begin{aligned}
\log q_1(\theta) = & -\exp\theta_t E_{U,S}((U_t - S_t)^2)/2 \\
& -\exp\theta_t E_U(\sum_i^N (U_t - V_{i,t})^2)/2 \\
& +(N+1)\log(\exp(\theta))/2 - \kappa'(\theta_t - \theta_{t-1})^2/2
\end{aligned}
\tag{35}
$$

### 2.2.4 Simplifying $q_1(S_t)$ and $\log q_1(U_t)$

Inspecting $\log q_1(S_t)$ and $\log q_1(U_t)$ we can see that both $q_1(S_t)$ and $q_1(U_t)$ are products of Normal distributions, and thus themselves Normal distributed

$$
q_1(S_t) \sim \mathcal{N}(S_t; \mu_{S,t}, 1/\tau_{S,t})
\tag{36}
$$

and

$$
q_1(U_t) \sim \mathcal{N}(U_t; \mu_{U,t}, 1/\tau_{U,t})
\tag{37}
$$

where

$$
\mu_{S,t} = (\lambda_S * 0 + \lambda_A A_t + E(\exp\theta_t)\mu_{U,t})/\tau_{S,t}
\tag{38}
$$

$$
\tau_{S,t} = \lambda_S + \lambda_A + E(\exp(\theta_t))
\tag{39}
$$

$$
\mu_{U,t} = (E_\theta(\exp(\theta_t)) \sum_i^N V_{i,t} + E_\theta(\exp(\theta_t))\mu_{S,t})/\tau_{U,t}
\tag{40}
$$

$$
\tau_{U,t} = (N+1) * E_\theta(\exp(\theta_t))
\tag{41}
$$

Note that $E_S(S_t) \approx \int q_1(S_t) S_t dS_t = \mu_{S,t}$ and $E_U(U_t) \approx \int q_1(U_t) U_t dU_t = \mu_{U,t}$

### 2.2.5 Simplifying $q_1(\theta_t)$

Regarding $q_1(\theta_t)$ we can expand a little using that

$$
\begin{aligned}
E_{U,S}((U_t - S_t)^2) &= E_{U,S}((U_t^2 + S_t^2 - 2S_t U_t)) \\
&= E_{U,S}(U_t^2) + E(S_t^2) - 2E(S_t)E(U_t) \\
&= \mu_U^2 + 1/\tau_U + \mu_S^2 + 1/\tau_S - 2\mu_S\mu_U \\
&= (\mu_U - \mu_S)^2 + 1/\tau_U + 1/\tau_S
\end{aligned}
\tag{42}
$$

(using that $E(X^2) = \mu^2 + 1/\tau$ for a normal distribution $\mathcal{N}(X; \mu, 1/\tau)$) and

$$
\begin{aligned}
E_U(\sum_i^N (U_t - V_{i,t})^2) &= E_U(\sum_i^N (U_t^2 + V_{i,t}^2 - 2U_t V_{i,t})) \\
&= E_U(\sum_i^N (U_t^2) + \sum_i^N (V_{i,t}^2) - 2\sum_i^N (U_t V_{i,t}) \\
&= E_U(N * (U_t^2) + \sum_i^N (V_{i,t}^2) - 2U_t \sum_i^N V_{i,t}) \\
&= N * (\mu_U^2 + 1/\tau_U) + \sum_i^N (V_{i,t}^2) - 2\mu_U \sum_i^N V_{i,t} = N/\tau_U + \sum_i^N (V_{i,t} - \mu_U)^2
\end{aligned}
\tag{43}
$$

Which together gives:

$$
\begin{aligned}
& E_{U,S}((U_t - S_t)^2) + E_U(\sum_i^N (U_t - V_{i,t})^2) \\
& = (\mu_U - \mu_S)^2 + 1/\tau_U + 1/\tau_S + (N+1)/\tau_U + \sum_i^N (V_{i,t} - \mu_U)^2
\end{aligned}
\tag{44}
$$

### 2.3 Approximating $q(\theta)$

We will approximate $q_1(\theta)$ with a Normal distribution.

To do this, we use a Laplace approximation around the max of $q_1(\theta)$ (see figure **Appendix 2—figure 2**): $\arg\max(q_1(\theta_t)) = \mu_{\theta,t}$ and with second derivative $-\tau_{\theta,t}$

This gives

$$q_1(\theta_t) \sim \mathcal{N}(\theta_t | \mu_{\theta,t}, 1/\tau_{\theta,t}) \tag{45}$$

### 2.3.1 First derivative

However, in order to find $\arg\max(q_1(\theta_t))$, we differentiate $\log q_1(\theta_t)$ and set equal to 0:

$$
\begin{aligned}
d\log q_1(\theta_t)/d\theta_t = &-\exp\theta_t E_{U,S}((U_t - S_t)^2)/2 \\
&-\exp\theta_t E_U(\sum_i^N (U_t - V_{i,t})^2)/2 \\
&+(N+1)/2 - \kappa'(\theta_t - \theta_{t-1}) = 0
\end{aligned} \tag{46}
$$

$$\exp\theta_t(E_{U,S}((U_t - S_t)^2)/2 + E_U(\sum_i^N (U_t - V_{i,t})^2)/2) = (N+1)/2 - \kappa'(\theta_t - \theta_{t-1}) \tag{47}$$

At this point there is no analytical solution.

### 2.3.2 Taylor expansion of first derivative

Although we could use a numerical approximation for speed of implementation, we use Taylor expansion. We need to solve for $\theta$

$$\exp\theta_t(E_{U,S}((U_t - S_t)^2)/2 + E_U(\sum_i^N (U_t - V_{i,t})^2)/2) - (N+1)/2 + \kappa'(\theta_t - \theta_{t-1}) = 0 \tag{48}$$

For simplicity we refer to $SV = (E_{U,S}((U_t - S_t)^2) + E_U(\sum_i^N (U_t - V_{i,t})^2))$

We can solve this by using the third-order Taylor expansion of the exponential:

$$
\begin{aligned}
\exp\theta_t \approx &\, e^{(\theta_*)} + e^{(\theta_*)}(\theta_t - \theta_*) + e^{(\theta_*)}(\theta_t - \theta_*)^2/2 + e^{(\theta_*)}(\theta_t - \theta_*)^3/6 = \\
e^{(\theta_*)}\big(1 - \theta_* &+ \theta_*^2/2 - \theta_*^3/6 + (1 - 2/2*\theta_* + 3/6*\theta_*^2)\theta_t + (1/2 - 3/6*\theta_*)\theta_t^2 + \theta_t^3/6\big) = \\
e^{(\theta_*)}\big(1 &- \theta_* + \theta_*^2/2 - \theta_*^3/6 + (1 - \theta_* + \theta_*^2/2)\theta_t + (1/2 - \theta_*/2)\theta_t^2 + \theta_t^3/6\big)
\end{aligned} \tag{49}
$$

We can set $\theta_*$ as $\theta_{t-1}$, as we expect that $\theta_t$ will be close to $\theta_*$. For big changes in the variance of $V_i$ this can be off; however, this was not a problem in this stimulus set which relied on slow gradual changes.

In order to find $\arg\max(q_1(\theta_t))$ we therefore have to solve

$$
\begin{aligned}
SV/2 * e^{(\theta_*)}\big(1 - \theta_* + \theta_*^2/2 - \theta_*^3/6 &+ (1 - \theta_* + \theta_*^2/2)\theta_t + (1/2 - \theta_*/2)\theta_t^2 + \theta_t^3/6\big) \\
&-(N+1)/2 + \kappa'(\theta_t - \theta_{t-1}) = 0
\end{aligned} \tag{50}
$$

which can be rewritten as a third order polynomial

$$\theta_t^3 + c_1\theta_t^2 + c_2\theta_t + c_3 = 0 \tag{51}$$

where

$$
\begin{aligned}
c_1 &= 1/2 - \theta_*/2 \\
c_2 &= 1 - \theta_* + \theta_*^2 + \kappa'/(SV/2 * e^{(\theta_*)}) \\
c_3 &= 1 - \theta_* + \theta_*^2/2 + \theta_*^3/6 - (N/2 + 1/2 + \kappa'\theta_{t-1})/(SV/2 * e^{(\theta_*)})
\end{aligned} \tag{52}
$$

the solution to which, $\theta_t^{optim}$, can be numerically found using Matlab's *nthroot* function.

As we assume that the log-variance only changes slightly between trials the solution closest to the previous value $\theta_{t-1}$ is automatically chosen, $\arg\max(q_1(\theta_t)) = \mu_{\theta_t} = \theta_t^{optim}$.

### 2.3.3 Second derivative

The second derivative is

$$d^2 \log q_1(\theta_t)/d\theta_t^2 = -\exp \theta_t * SV/2 - \kappa' \tag{53}$$

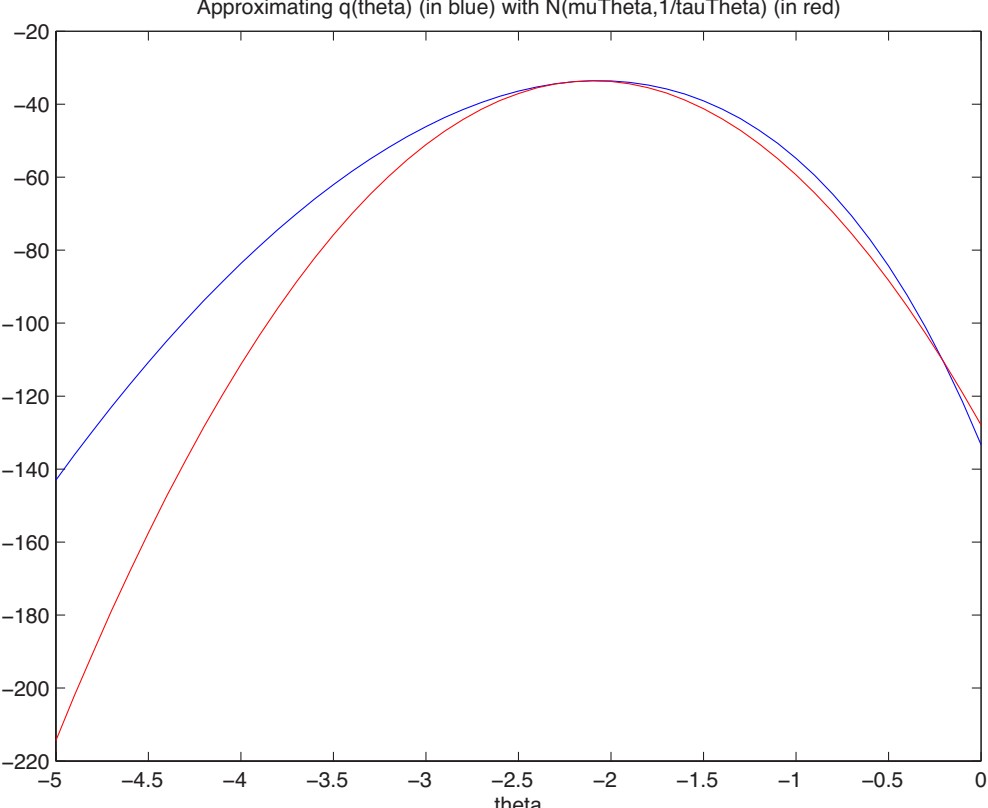

**Appendix 2—figure 2.** Approximation of theta using Laplace approximation.

We evaluate this at $argmax(q_1(\theta_t))$, so we insert $\theta_t = \mu_{\theta,t}$

Hence we can finally write

$$q_1(\theta_t) \sim \mathcal{N}(\theta_t | \mu_{\theta,t}, 1/\tau_{theta,t}) \tag{54}$$

where

$$\mu_{\theta,t} = \theta_t^{optim} \tag{55}$$

$$\tau_{\theta,t} = \exp(\mu_{\theta,t}) * (SV)/2 + \kappa' \tag{56}$$

and where $SV = (\mu_U - \mu_S)^2 + 1/\tau_S + (N+1)/\tau_U + \sum_i^N (V_{i,t} - \mu_U)^2$

With $q_1(\theta_t)$ a Normal distribution, that makes $q_1(\lambda_{V,t})$ a log-normal distribution with $\mu_{\lambda_{V,t}} = E(\lambda_{V,t}) = E(\exp(\theta_t)) = \exp(\mu_{\theta_t} + 1/(2 * \tau_{\theta_t}))$ (general property of log-normal distribution).

## 2.4 Final algorithm for C = 1

We can now create an iterative algorithm that for each time step $t$ represents the model posterior. Variables $\sigma_A^2 = 1/\lambda_A$, $\sigma_0^2 = 1/\lambda_0$ and $\kappa$ have to be set before hand, together with the input data $A_{1:t}$ and $V_{1:N,1:t}$. For time step $t$:

1. initially set

$$\mu_{\theta,t} = \mu_{\theta,t-1}, \tag{57}$$

$$\mu_{S,t} = 0, \tag{58}$$

$$\mu_{U,t} = 1/N \sum V_{i,t}, \tag{59}$$

$$\tau_{\theta,t} = 1 \tag{60}$$

2. set $\mu_S, \tau_{S,t}$

$$\mu_{S,t} = (\lambda_A A_t + \exp(\mu_{\theta_t} + 1/(2*\tau_{\theta_t}))\mu_{U,t})/\tau_{S,t} \tag{61}$$

$$\tau_{S,t} = \lambda_S + \lambda_A + \exp(\mu_{\theta_t} + 1/(2*\tau_{\theta_t})) \tag{62}$$

3. set $\mu_{U,t}, \tau_{U,t}$

$$\mu_{U,t} = (N/(N+1))\bar{V}_t + (1/(N+1))\mu_{S,t} \tag{63}$$

$$\tau_{U,t} = (N+1)*\exp(\mu_{\theta_t} + 1/(2*\tau_{\theta_t})) \tag{64}$$

where $\bar{V}_t = 1/N \sum_i^N V_{i,t}$
  4. find $\mu_{\theta_t}$ by solving third order polynomial, *Equation 51*,

$$\mu_{\theta,t} = \theta_{optim} \tag{65}$$

then set $\tau_{\theta_t}$

$$\tau_{\theta,t} = k' + \exp(\mu_{\theta,t})*((\mu_{U,t} - \mu_{S,t})^2 + 1/\tau_{S,t} + (N+1)/\tau_{U,t} + \sum_i^N (V_{i,t} - \mu_{U,t})^2)/2 \tag{66}$$

where $\kappa'_t = 1/(1/\kappa + 1/\tau_{\theta,t-1})$
  5. Repeat steps 2–4 until the change in each parameter is small (<0.0001)
  This is then repeated for each time step $t$, providing us with the approximation to the posterior
$P_1(S_t, \theta_t, U_t | A_t, V_{1:N,t}) \approx q_1(S_t, U_t, \theta_t)) = q_1(S_t) * q_1(U_t) * q_1(\theta_t)$.

## 2.5 Posterior for C = 2

Due to the independent structure this posterior can be written as

$$P_2(S_{A,t}, S_{V,t}, \lambda_{V,t}, U_t | A_t, V_{1:N,t}) = P(S_{A,t}|A_t)P(S_{V,t}, \lambda_{V,t}, U_t | V_{1:N,t}) \tag{67}$$

where

$$P_2(S_{A,t}|A_t) = P(S_{A,t})P(A_t|S_{A,t})/Z \tag{68}$$

which is simple enough given the Normal distribution of both $P(S_{A,t})$ and $P(A_t|S_{A,t})$

$$P_2(S_{A,t}|A_t) = \mathcal{N}(S_{A,t}; A_t\sigma_{A0}^2/\sigma_A^2, \sigma_{A0}^2) \tag{69}$$

where $\sigma_{A0}^2 = 1/(1/\sigma_A^2 + 1/\sigma_0^2)$.
  Note that for the subject response the posterior $P_2(S_{A,t}|A_t)$ is all that is needed, but for the calculation of the prior $P(\lambda_{V,t})$ for subsequent trial $t+1$ we need to compute the full posterior.
  We again use the transformation of parameters

$$\theta_t = \log(\lambda_{V,t}) \tag{70}$$

Proceeding with just the posterior over $S_{V,t}$, $U_t$ and $\theta_t$

$$P_{C=2}(S_{V,t}, \theta_t, U_t | A_{1:t}, V_{1:N,1:t}) \propto P(S_{V,t})P(U_t|S_{V,t}, \theta_t)P(V_{1:N,t}|U_t, \theta_t) \\ \int P(\theta_t|\theta_{V,t-1})P(\theta_{V,1:t-1}|A_{1:t-1}, V_{1:N,1:t-1})d\theta_{V,1:t-1}/Z \tag{71}$$

where

$$P(U_t|S_{V,t}, \theta_t) = \mathcal{N}(U_t; S_{V,t}, 1/\exp(\theta_t)) \tag{72}$$

$$P(V_{1:N,t}|U_t, \theta_t) = \prod_n \mathcal{N}(U_t; V_{n,t}, 1/\exp(\theta_t)) \tag{73}$$

$$P(\theta_t|\theta_{t-1}) = \mathcal{N}(\theta_t; \theta_{t-1}, 1/\kappa) \tag{74}$$

We will assume that $P(\theta_{V,1:t-1}|A_{1:t-1}, V_{1:N,1:t-1})$ can be approximated by a Normal distribution (see $q_1(\theta)$ below), thus allowing us to use properties of Normal distributions.

Hence,

$$\int P(\theta_t|\theta_{t-1})P(\theta_{V,1:t-1}|A_{1:t-1}, V_{1:N,1:t-1})\, d\theta_{V,1:t-1} = \mathcal{N}(\theta_t; \theta_{t-1}, 1/\kappa'_t) \tag{75}$$

where $1/\kappa'_t = 1/\kappa + 1/\tau_{\theta,t-1}$ (due to the convolution of $P(\theta_t|\theta_{t-1})$ with $P(\theta_{t-1}|A_{1:t-1}, V_{1:N,1:t-1})$, both Normal distributed).

While any estimate of $\theta_t$ will depend on $A_{1:t-1}$ and $V_{1:N,1:t-1}$) for ease of notation, we will omit those in the following.

The log-posterior is now

$$\begin{aligned}
\log P_2(S_{V,t}, \theta_t, U_t|A_{1:t}, V_{1:N,1:t}) &\propto \\
&-\lambda_0 S_{V,t}^2/2 \\
&-\exp\theta_t(U_t - S_{V,t})^2/2 + \theta_t/2 \\
&-\exp\theta_t \sum_i^N (U_t - V_{i,t})^2/2 + N\theta_t/2 \\
&-\kappa'(\theta_t - \theta_{V,t-1})^2/2
\end{aligned} \tag{76}$$

## 2.6 Variational Bayes approximation for C = 2

We will now approximate the log-posterior with variational Bayes by factorization: $P_2(S_t, \theta_t, U_t|A_t, V_{1:N,t}) \approx q_2(S_t, U_t, \theta_t)) = q_2(S_t) * q_2(U_t) * q_2(\theta_t)$ This proceeds similarly to the combined ($C = 1$) model, but with $S_{V,t}$ instead of $S_t$, and with no influence from $A_t$. For completeness the calculations are included here:

### 2.6.1 $q_2(S_t)$

For $q_2(S_t)$

$$\log q_2(S_{V,t}) \propto -\lambda_0 S_{V,t}^2/2 - E_\theta(\exp(\theta_t))E_U((U_t - S_{V,t})^2)/2 \tag{77}$$

where $E_Y(X)$ signifies the expectation of X over the distribution of Y: $E_Y(X) = \int P(Y)X dY$.

Using $\quad E_U((U_t - V, t)^2) = E_U(U_t^2 + S_{V,t}^2 - 2U_t S_{V,t}) = E_U(U_t^2) + S_{V,t}^2 - 2S_{V,t}E_U(U_t) \quad =$ $(S_{V,t}^2 - S_{V,t}E_U(U_t) * 2 + E_U(U_t)^2) - E_U(U_t)^2 + E_U(U_t^2) = (S_{V,t} - E_U(U_t))^2 - E_U(U_t)^2 + E_U(U_t^2)$, where the last two terms do not depend on $S_{V,t}$ (and thus can be discarded) we can rewrite the last term:

$$\log q_2(S_{V,t}) \propto -\lambda_0 S_{V,t}^2/2 - E_\theta(\exp(\theta_t))(S_{V,t} - E_U(U_t))^2/2 \tag{78}$$

### 2.6.2 $q_2(U_t)$

For $q_2(U_t)$

$$\log q_2(U_t) \propto -E_\theta(\exp(\theta_t))/2 \sum_i (U_t - V_{i,t})^2 - E_\theta(\exp(\theta_t))E_S((U_t - S_{V,t})^2)/2 \tag{79}$$

Here, we use the same trick

$$\log q_2(U_t) \propto -E_{\theta_t}(\exp(\theta_t))/2 \sum_i (U_t - V_{i,t})^2 - E_\theta(\exp(\theta_t))(U_t - E_S(S_{V,t}))^2)/2 \tag{80}$$

### 2.6.3 $q_2(\theta_t)$

For $q_2(\theta)$

$$\begin{aligned}
\log q_2(\theta) = & -\exp\theta_t E_{U,S}((U_t - S_{V,t})^2)/2 \\
& -\exp\theta_t E_U(\sum_i^N (U_t - V_{i,t})^2)/2 \\
& +(N+1)\log(\exp(\theta))/2 - \kappa'(\theta_t - \theta_{t-1})^2/2
\end{aligned} \tag{81}$$

## 2.6.4 Simplifying $q_2(S_t)$ and $\log q_2(U_t)$

Inspecting $\log q_2(S_{V,t})$ and $\log q_2(U_t)$ we can see that both $q_2(S_{V,t})$ and $q_2(U_t)$ are products of Normal distributions, and thus themselves Normal distributed

$$q_2(S_{V,t}) \sim \mathcal{N}(S_{V,t}|\mu_{S_V,t}, 1/\tau_{S_V,t}) \tag{82}$$

and

$$q_2(U_t) \sim \mathcal{N}(U_t|\mu_{U,t}, 1/\tau_{U,t}) \tag{83}$$

where

$$\mu_{S_V,t} = (\lambda_0 * 0 + E(\exp(\theta_t))\mu_{U,t})/\tau_{S_V,t} \tag{84}$$

$$\tau_{S_V,t} = \lambda_0 + E(\exp(\theta_t)) \tag{85}$$

$$\mu_{U,t} = (E_\theta(\exp(\theta_t)) \sum_i^N V_{i,t} + E_\theta(\exp(\theta_t))\mu_{S_V,t})/\tau_{U,t} \tag{86}$$

$$\tau_{U,t} = (N+1) * E_\theta(\exp(\theta_t)) \tag{87}$$

Note that $E_S(S_{V,t}) \approx \int q_2(S_{V,t})S_{V,t}dS_{V,t} = \mu_{S_V,t}$ and $E_U(U_t) \approx \int q_2(U_t)U_t dU_t = \mu_{U,t}$

We can approximate $q_2(\theta)$ with a Normal distribution, in exactly the same way as for C = 1. As equations are identical (see above) they will not be repeated here.

## 2.7 Final algorithm for C = 2

We can now create an iterative algorithm that for each time step $t$ represents the variational Bayes approximation of the model posterior over $S_{V,t}$, $U_t$ and $\lambda_{V,t}$ (or rather $\theta_t$):, $P(S_{V,t}, U_t, \theta_t|V_{1:N,t})$. Variable $\kappa$ has to be set before hand, together with the input data $A_{1:t}$ and $V_{1:N,1:t}$. For time step $t$:

1. initially set $\mu_{\theta,t} = \mu_{\theta,t-1}$, $\mu_{U,t} = 1/N \sum V_{i,t}$, and $\tau_{\theta,t} = 1$
2. set $\mu_{S_V}$, $\tau_{S_V,t}$

$$\mu_{S_V,t} = (\exp(\mu_{\theta_t} + 1/(2*\tau_{\theta,t}))\mu_{U,t})/\tau_{S_V,t} \tag{88}$$

$$\tau_{S_V,t} = \lambda_0 + \exp(\mu_{\theta_t} + 1/(2*\tau_{\theta,t})) \tag{89}$$

3. set $\mu_U$, $\tau_{U,t}$

$$\mu_{U,t} = (N\bar{V}_t + \mu_{S_V,t})/(N+1) \tag{90}$$

$$\tau_{U,t} = (N+1) * \exp(\mu_{\theta_t} + 1/(2*\tau_{\theta,t})) \tag{91}$$

where $\bar{V}_t = 1/N \sum_i^N V_{i,t}$

4. find $\mu_{\theta,t}$ by numerically solving polynomial, *Equation 51*,

$$\mu_{\theta,t} = \theta_{optim} \tag{92}$$

then set $\tau_{\theta,t}$

$$\tau_{\theta,t} = k' + \exp(\mu_{\theta,t}) * ((\mu_{U,t} - \mu_{S_V,t})^2 + 1/\tau_{S_V,t} + (N+1)/\tau_{U,t} + \sum_{i}^{N}(V_{i,t} - \mu_{U,t})^2)/2 \tag{93}$$

where $\kappa'_t = 1/(1/\kappa + 1/\tau_{\theta,t-1})$

5. Repeat steps 2–4 until convergence, that is, until the change in each parameter is small (<0.0001)

This is then repeated for each time step $t$, providing us with the approximation to the posterior $P_2(S_t, \theta_t, U_t | A_t, V_{1:N,t}) \approx q_2(S_t, U_t, \theta_t)) = q_2(S_t) * q_2(U_t) * q_2(\theta_t)$.

See figure *Appendix 2—figure 3* below for an example of the learned inference of the visual variance $\sigma_{V,t}^2 = 1/\lambda_{V,t} \sim 1/\log q_2(\theta_t)$, compared with a simple instantaneous learner model that assumes $\sigma_{V,t}^2 = 1/N \sum_i (V_{i,t} - \bar{V}_t)^2$.

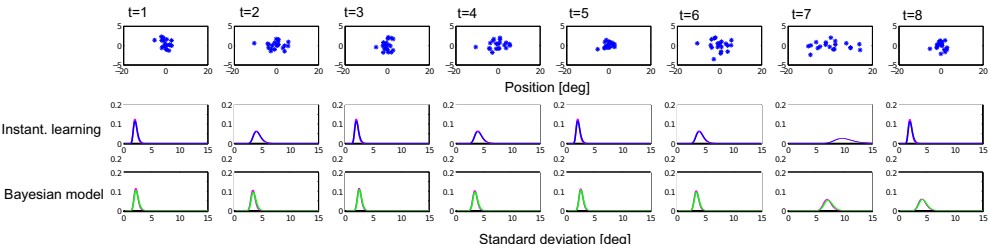

**Appendix 2—figure 3.** Comparing variational Bayes approximation with a numerical discretised grid approximation. Top row: Example visual stimuli over eight subsequent trials. Middle row: The distribution of estimated sample variance, with no learning over trials. Bottom row: The distribution of _V;t for the Bayesian model that incorporates the learning across trials. Red line is the numerical comparison when using a discretised grid to estimate variance, as opposed to the variational Bayes (green line).

## 3 Marginal model evidence

Recall that the posterior is a mixture of the forced fusion and independent solutions, with the mixture determined by the posterior probability of either model generating the data:

$$P(C_t = 1 | A_t, V_{1:N,t}) = \frac{P(A_t, V_{1:N,t} | C_t = 1)P(C_t = 1)}{P(A_t, V_{1:N,t} | C_t = 1)P(C_t = 1) + P(A_t, V_{1:N,t} | C_t = 2)P(C_t = 2)} \tag{94}$$

To evaluate this, we need to calculate the marginal model evidence, $P(A_t, V_{1:N,t} | C_t)$, for either model.

One way to do so is by a sampling approximation, but here we utilise the variational results we have already found.

### 3.1 Model likelihood for C = 2, two sources $S_{V,t}, S_{A,t}$

We need to evaluate the model likelihood for both $C = 1$ and $C = 2$. The case for $C = 2$ is slightly simpler, hence we start with this:

$$P(A_t, V_{1:N,t} | C_t = 2) = P(A_t | C_t)P(V_{1:N,t} | C_t = 2) =$$
$$\int P(A_t | S_{A,t}, C_t = 2)P(S_{A,t} | C_t = 2)dS_{A,t} *$$
$$\int P(V_{1:N,t} | U_t, \lambda_t, C_t = 2)P(U_t, \lambda_t, S_{V,t} | C_t = 2)dVd\lambda dS_{V,t} \tag{95}$$

The first integral is easy as it is just the integral of the product of two Normal distributions.

$$\int P_2(A_t|S_{A,t})P_2(S_{A,t})dS = \frac{1}{\sqrt{(2\pi(\sigma_A^2+1/\tau_0))}}\exp(-\frac{(A-\mu_0)^2}{2(\sigma_A^2+1/\tau_0)}) \tag{96}$$

It is however more convenient to operate in log-space

$$\log P_2(A_t) = -\log(\sqrt{2\pi(\sigma_A^2+1/\tau_0)}) - (A-\mu_0)^2/(2(\sigma_A^2+1/\tau_0)) \tag{97}$$

The second integral we approximate through the Free Energy that we already maximise iteratively in the variational Bayes algorithm.

$$\log P(V_{1:N,t}|C_t=2) = \log \int P(V_{1:N,t}|U_t,\theta_t,S_{V,t},C_t=2)P(U_t,\theta_t,S_{V,t}|C_t=2)dU_t d\theta_t dS_{V,t}$$

$$\approx L_2(q) = \int q_2(U_t,\theta_t,S_{V,t})* \tag{98}$$

$$\log\frac{P_2(V_{1:N,t}|U_t,\theta_t,C_t)P_2(U_t|,S_{V,t},\theta_t,C_t)P_2(S_{V,t}|C_t)P_2(\theta_t|C_t)}{q_2(U_t,\theta_t,S_{V,t})}dU_t d\theta_t dS_{V,t}$$

(this approximation becomes exact if the variational approximation is exact, that is, if the Kulback-Leibler difference between the posterior $P_2(U_t,\theta_t,S_{V,t}|V_{1:N})$ and the approximation $q_2(U_t,\theta_t,S_{V,t})$ becomes zero.)

This can be interpreted as taking the expectation with regard to the posterior approximation, and due to the properties of the logarithm this can be separated into a sum of expectations:

$$L_2(q) = E(\log P_2(V_{1:N,t}|U_t,\theta_t))+$$
$$E(\log P_2(U_t|S_{V,t},\theta_t)) + E(\log P_2(S_{V,t})) + E(\log P_2(\theta_t)) \tag{99}$$
$$-E(\log q_2(U_t)) - E(\log q_2(\theta_t)) - E(\log q_2(S_{V,t}))$$

where (due to *Equation 43*)

$$E(\log P_2(V_{1:N,t}|U_t,\theta_t)) = E(\log \prod_i P_2(V_{i,t}|U_t,\theta)) =$$
$$E(\log \prod_i \sqrt{\tfrac{\exp(\theta_t)}{2\pi}}\exp(-(V_{i,t}-U_t)^2\exp(\theta_t)/2) =$$
$$E(\sum_i \log(\sqrt{\tfrac{\exp(\theta_t)}{2\pi}}) - (V_{i,t}-U_t)^2\exp(\theta_t)/2) = \tag{100}$$
$$N/2(E_{\theta_t}(\theta) - \log(2\pi)) + E_{U_t}(\sum_i -(V_{i,t}-U_t)^2))E(\exp(\theta_t))/2 =$$
$$N/2(\mu_\theta - \log(2\pi)) - (N/\tau_{U,t} + \sum_i(V_{i,t}-\mu_U)^2)\exp(\mu_{\theta,t}+1/(2\tau_{\theta,t}))/2$$

and (since $E(X^2) = \mu_X^2 + \sigma_X^2$)

$$E(\log P_2(U_t)|S_{V,t},\theta_t)) = E(\log \mathcal{N}(U_t;S_{V,t},1/\exp(\theta_t))) =$$
$$E(\log \sqrt{\tfrac{\exp\theta_t}{2\pi}}\exp(-(U_t-S_{V,t})^2)\exp(\theta_t)/2)) = \tag{101}$$
$$(\mu_{\theta,t} - \log(2\pi))/2 - [(\mu_U-\mu_{Sv,t})^2 + 1/\tau_{U,t} + 1/\tau_{Sv,t}]\exp(\mu_{\theta,t}+1/(2\tau_{\theta,t}))/2$$

and

$$E(\log P_2(S_{V,t})) = E(\log \mathcal{N}(S_{V,t};\mu_0,\sigma_0^2)) =$$
$$E(\log \sqrt{\tfrac{1}{\sigma_0^2 2\pi}}\exp(-(S_{V,t}-\mu_0)^2)/(2\sigma_0^2)) = -\log(\sigma_0^2 2\pi)/2 - [(\mu_{Sv,t}-\mu_0)^2 + 1/\tau_{Sv,t}]/(2\sigma_0^2) \tag{102}$$

and (due to *Equation 75*)

$$E(\log P_2(\theta_t)) = E(\mathcal{N}(\theta_t;\theta_{t-1},1/\kappa')) =$$
$$E(\log \sqrt{\tfrac{\kappa'}{2\pi}}\exp(-(\theta_t-\mu_{\theta,t-1})^2\kappa'/2)) = (\log\kappa' - \log(2\pi))/2 - [(\mu_{\theta,t}-\mu_{\theta,t-1})^2 + 1/\tau_{\theta,t}]\kappa'/2 \tag{103}$$

and

$$E(\log q_2(U_t)) = E(\log \mathcal{N}(U_t; \mu_{U,t}, 1/\tau_{U,t})) =$$
$$E(\log(\sqrt{\tfrac{\tau_{U,t}}{2\pi}} \exp(-(U_t - \mu_{U,t})^2 \tau_{U,t}/2))) =$$
$$\log \sqrt{\tfrac{\tau_{U,t}}{2\pi}} - E((U_t - \mu_{U,t})^2 \tau_{U,t}/2) =$$
$$(\log \tau_{U,t} - \log(2\pi))/2 - (E(U_t^2) + \mu_{U,t}^2 - 2\mu_{U,t}E(U_t))\tau_{U,t}/2 =$$
$$(\log \tau_{U,t} - \log(2\pi))/2 - (\mu_{U,t}^2 + 1/\tau_{U,t}^2 + \mu_{U,t}^2 - 2\mu_{U,t}\mu_{U,t})\tau_{U,t}/2 =$$
$$(\log \tau_{U,t} - \log(2\pi) - 1)/2 \tag{104}$$

and

$$E(\log q_2(S_t)) = E(\log \mathcal{N}(S_{V,t}; \mu_{S_V,t}, 1/\tau_{S_V,t})) =$$
$$E(\log \sqrt{\tfrac{1}{2\pi\sigma_0^2}} \exp(-(S_{V,t} - \mu_{S_V,t})^2 \tau_{S_V,t}/2)) =$$
$$(\log \tau_{S_V,t} - \log(2\pi) - 1)/2 \tag{105}$$

and

$$E(\log q_2(\theta_t)) = E(\log \mathcal{N}(\theta_t; \mu_{\theta,t}, 1/\tau_{\theta,t})) =$$
$$(\log \tau_{\theta,t} - \log(2\pi) - 1)/2 \tag{106}$$

In total we now have

$$\log P_2(A_t, V_{1:N,t}|C_t) \approx \log P_2(A_t) + L_2 =$$
$$-(\log(2\pi) + \log(\sigma_A^2 + 1/\tau_0))/2 - (A - \mu_0)^2/(2(\sigma_A^2 + 1/\tau_0))$$
$$+(\mu_{\theta,t} - \log(2\pi))N/2 - [N/\tau_{U,t} + \sum_i^N (V_{i,t} - \mu_U)^2]\exp(\mu_{\theta,t} + 1/(2\tau_{\theta,t}))/2$$
$$+((\mu_{\theta,t} - \log(2\pi))/2 - [(\mu_U - \mu_{S_V,t})^2 + 1/\tau_{U,t} + 1/\tau_{S_V,t}]\exp(\mu_{\theta,t} + 1/(2\tau_{\theta,t}))/2$$
$$-\log(\sigma_0^2 2\pi)/2 - [(\mu_{S_V,t} - \mu_0)^2 + 1/\tau_{S_V,t}]/(2\sigma_0^2)$$
$$+(\log \kappa' - \log(2\pi))/2 - [(\mu_{\theta,t} - \mu_{\theta,t-1})^2 + 1/\tau_{\theta,t}]\kappa'/2$$
$$-(\log \tau_{U,t} - \log(2\pi) - 1)/2$$
$$-(\log \tau_{S_V,t} - \log(2\pi) - 1)/2$$
$$-(\log \tau_{\theta,t} - \log(2\pi) - 1)/2 \tag{107}$$

Although lengthy, this is trivial and fast to compute numerically in Matlab (e.g.). Note that all estimates come from the variational Bayes approximation $q_2(S_t, U_t, \theta_t)$.

## 3.2 Model likelihood for C = 1, one source $S_t = S_{V,t} = S_{A,t}$

We now need to do the same for the one source model.

$$P(A_t, V_{1:N,t}|C_t = 1) = P_1(A_t, V_{1:N,t}) =$$
$$\int P_1(A_t|S_t, C_t = 1)P_1(V_{1:N,t}|U_t, \lambda_t, C_t = 1)P_1(U_t, \lambda_t, S_t|C_t = 1)dVd\lambda dS_t \tag{108}$$

Note that for simplicity in notation we will use $P_1$ to indicate the probability within the model given $C_t = 1$

We again approximate through the Free Energy that we already maximised iteratively in the variational Bayes algorithm.

$$\log P_1(A_t, V_{1:N,t}) = \log \int P_1(V_{1:N,t}|U_t, \theta_t, S_t)P_1(A_t|S_t)P_1(U_t, \theta_t, S_t)dU_t d\theta_t dS_t$$
$$\approx L_{C_t=1}(q_1) =$$
$$\int q_1(U_t, \theta_t, S_{V,t}) \log \frac{P_1(V_{1:N,t}|U_t, \theta_t)P_1(A_t|S_t)P_1(U_t|S_t, \theta_t)P_1(S_t)P_1(\theta_t)}{q_1(U_t, \theta_t, S_t)}dU_t d\theta_t dS_t \tag{109}$$

(this approximation becomes exact if the variational approximation is exact, ie if the Kulback-Leibler difference between the posterior $P_1(V, \theta_t, S_{V,t}|V_{1:N,t})$ and the approximation $q_1(U_t, \theta_t, S_{V,t})$ becomes zero.)

This can be interpreted as taking the expectation with regard to the posterior approximation, and due to the properties of the logarithm this can be separated into a sum of expectations:

$$L_1(q) = E(\log P_1(A_t|S_t)) + E(\log P_1(V_{1:N,t}|U_t, \lambda_t))$$
$$+ E(\log P_1(U_t)) + E(\log P_1(\theta_t)) + E(\log P_1(S_{V,t}))$$
$$- E(\log q_1(U_t)) - E(\log q_1(\theta_t)) - E(\log q_1(S_{V,t})) \quad (110)$$

where (since $E(X^2) = \mu_X^2 + \sigma_X^2$)

$$E(\log P_1(A_t|S_t)) =$$
$$E\left(\log \sqrt{\frac{1}{2\pi\sigma_A^2}} \exp\left(-(A_t - S_t)^2/(2\sigma_A^2)\right)\right) = \quad (111)$$
$$-\log(\sigma_A^2 2\pi)/2 - [(A_t - \mu_S)^2 + 1/\tau_{S,t}]/(2\sigma_A^2)$$

$$E(\log P_1(V_{1:N,t}|U_t, \theta_t)) = E(\log \prod_i P_1(V_{i,t}|U_t, \theta)) =$$
$$E(\log \prod_i \sqrt{\frac{\exp(\theta_t)}{2\pi}} \exp(-(V_{i,t} - U_t)^2 \exp(\theta_t)/2) =$$
$$E(\sum_i \log(\sqrt{\frac{\exp(\theta_t)}{2\pi}}) - (V_{i,t} - U_t)^2 \exp(\theta_t)/2) = \quad (112)$$
$$n/2(E_{\theta_t}(\theta) - \log(2\pi)) + E(\exp(\theta_t))/2E_{U_t}(\sum_i -(V_{i,t} - U_t)^2)) =$$
$$n/2(\mu_{\theta,t} - \log(2\pi)) + \exp(\mu_{\theta,t} - 1/(2\tau_{\theta,t}))/2(N/\tau_{U,t} + \sum_i (V_{i,t} - \mu_U)^2)$$

and

$$E(\log P_1(U_t)|S_t, \theta_t)) = E(\log \mathcal{N}(U_t; S_t, 1/exp(\theta_t))) =$$
$$E(\log \sqrt{\frac{\exp\theta_t}{2\pi}} \exp(-(U_t - S_t)^2) \exp(\theta_t)/2)) = \quad (113)$$
$$(\mu_{\theta,t} - \log(2\pi))/2 - [(\mu_U - \mu_{S,t})^2 + 1/\tau_{U,t} + 1/\tau_{S,t}] \exp(\mu_{\theta,t} + 1/(2\tau_{\theta,t}))/2$$

and

$$E(\log P_1(S_t)) = E(\log \mathcal{N}(S_t; \mu_0, \sigma_0^2)) =$$
$$E(\log \sqrt{\frac{1}{\sigma_0^2 2\pi}} \exp(-(S_t - \mu_0)^2)/(2\sigma_0^2)) = -\log(\sigma_0^2 2\pi)/2 - [(\mu_{S,t} - \mu_0)^2 + 1/\tau_{S,t}]/(2\sigma_0^2) \quad (114)$$

and (due to *Equation 75*)

$$E(\log P_1(\theta_t)) = E(\mathcal{N}(\theta_t; \theta_{t-1}, 1/\kappa')) =$$
$$E(\log \sqrt{\frac{\kappa'}{2\pi}} \exp(-(\theta_t - \mu_{\theta,t-1})^2 \kappa'/2)) = (\log \kappa' - \log(2\pi))/2 - [(\mu_{\theta,t} - \mu_{\theta,t-1})^2 + 1/\tau_{\theta,t}]\kappa'/2 \quad (115)$$

and

$$E(\log q_1(U_t)) = E(\log \mathcal{N}(U_t; \mu_{U,t}, 1/\tau_{U,t})) =$$
$$E(\log \sqrt{\frac{\tau_{U,t}}{2\pi}} \exp(-(U_t - \mu_{U,t})^2 \tau_{U,t}/2)) =$$
$$\log \sqrt{\frac{\tau_{U,t}}{2\pi}} - E((U_t - \mu_{U,t})^2 \tau_{U,t}/2) = \quad (116)$$
$$(\log \tau_{U,t} - \log(2\pi))/2 - (E(U_t^2) + \mu_{U,t}^2 - 2\mu_{U,t}E(U_t))\tau_{U,t}/2 =$$
$$(\log \tau_{U,t} - \log(2\pi) - 1)/2$$

and

$$E(\log q_1(S_t)) = E(\log \mathcal{N}(S_t; \mu_{S,t}, 1/\tau_{S,t})) =$$
$$E(\log \sqrt{\frac{1}{2\pi\sigma_0^2}} \exp(-(S_t - \mu_{S,t})^2 \tau_{S,t}/2)) = \quad (117)$$
$$(\log \tau_{S,t} - \log(2\pi) - 1)/2$$

and

$$E(\log q_1(\theta_t)) = E(\log \mathcal{N}(\theta_t; \mu_{\theta,t}, 1/\tau_{\theta,t})) =$$
$$(\log \tau_{\theta,t} - \log(2\pi) - 1)/2 \quad (118)$$

In total we now have

$$\log P_1(A_t, V_{1:N,t}) \approx$$
$$L_1(q_1) = -\log(\sigma_A^2 2\pi)/2 - [(A_t - \mu_S)^2 + 1/\tau_{S,t}]/(2\sigma_A^2)$$
$$+ N/2(\mu_{\theta,t} - \log(2\pi)) - (N/\tau_{U,t} + \sum_i^N (V_{i,t} - \mu_U)^2)\exp(\mu_{\theta,t} + 1/(2\tau_{\theta,t}))/2$$
$$+ ((\mu_{\theta,t} - \log(2\pi))/2 - [(\mu_U - \mu_{S_V,t})^2 + 1/\tau_{U,t} + 1/\tau_{S_V,t}]\exp(\mu_{\theta,t} + 1/(2\tau_{\theta,t}))/2$$
$$- \log(\sigma_0^2 2\pi)/2 - [(\mu_{S_V,t} - \mu_0)^2 + 1/\tau_{S_V,t}]/(2\sigma_0^2)$$
$$+ (\log\kappa' - \log(2\pi))/2 - [(\mu_{\theta,t} - \mu_{\theta,t-1})^2 + 1/\tau_{\theta,t}]\kappa'/2$$
$$- (\log\tau_{U,t} - \log(2\pi) - 1)/2$$
$$- (\log\tau_{S,t} - \log(2\pi) - 1)/2$$
$$- (\log\tau_{\theta,t} - \log(2\pi) - 1)/2$$

(119)

(where all first and second order moments $(\mu, \tau)$ have been derived from $q_1$).

This is identical to the result from $C = 2$ except for the first line.

In total this provides us with an approximation to the model evidence for each model, $P(A_t, V_{1:N,t}|C_t = 1)$ and $P(A_t, V_{1:N,t}|C_t = 2)$, which can be used to calculate the posterior probability of either model given data, $P(C_t|A_{1:t}, V_{1:N,1:t})$.

## 4 Putting it all together

For either sub-model, the factorization (due to assumptions and variational Bayes approximation) allows us to write out the equations for the variable for subject choice:

$$P(S_t|A_{1:t}, V_{1:N,1:t}) =$$
$$P(S_t|A_{1:t}, V_{1:N,1:t}, C_t = 1)P(C_t = 1|A_{1:t}, V_{1:N,1:t}) +$$
$$P(S_{A,t}|A_{1:t}, V_{1:N,1:t}, C_t = 2)P(C_t = 2|A_{1:t}, V_{1:N,1:t}) \approx$$
$$q_{C=1,t}(S_t)P(C_t = 1|A_{1:t}, V_{1:N,1:t}) +$$
$$P(S_{A,t}|A_{1:t}, C_t = 2)P(C_t = 2|A_{1:t}, V_{1:N,1:t})$$

(120)

This is now a mixture of two Gaussian distributions (due to the Variational Bayes approximation), with mixture weights given by the model evidence (partly approximated by the Free Energy).

We will assume subjects report the mean of the distribution that is,

$$\hat{S}_t = \hat{S}_{C=1,t}P(C_t = 1|A_{1:t}, V_{1:N,1:t}) + \hat{S}_{C=2,t}P(C_t = 2|A_{1:t}, V_{1:N,1:t})$$

(121)

where $\hat{S}_{C=1,t} = \mu_{S,t}$ for $C = 1$ and $\hat{S}_{C=2,t} = \mu_{S_A,t}$ for $C = 2$.

We also need a prior over the visual log-reliability for the following trial

$$P(\theta_t|A_{1:t}, V_{1:N,1:t}) =$$
$$P(\theta_t|A_{1:t}, V_{1:N,1:t}, C = 1)P(C = 1|A_{1:t}, V_{1:N,1:t}) +$$
$$P(\theta_t|A_{1:t}, V_{1:N,1:t}, C = 2)P(C = 2|A_{1:t}, V_{1:N,1:t})$$

(122)

While this is a mixture of two Gaussians, we need the prior to be a single Gaussian in order for our approximation scheme above to work. We will approximate this mixture with a single Gaussian (essentially fitting a Gaussian to the mixture of two Gaussians).

$$P(\theta_t|A_{1:t}, V_{1:N,1:t}) \approx \mathcal{N}(\theta_t|\mu_{\theta,t}, 1/\tau_{\theta,t})$$

(123)

where (due to the first- and second-order moments of mixture distributions)

$$\mu_{\theta,t} = \mu_{\theta,t,C=1}P(C_t = 1|A_{1:t}, V_{1:N,1:t}) + \mu_{\theta,t,C=2}P(C_t = 2|A_{1:t}, V_{1:N,1:t})$$

(124)

$$1/\tau_{\theta,t} = (1/\tau_{\theta,t,C=1} + \mu_{\theta,t,C=1}^2)P(C_t = 1|A_{1:t}, V_{1:N,1:t})$$
$$+ (1/\tau_{\theta,t,C=2} + \mu_{\theta,t,C=2}^2)P(C_t = 2|A_{1:t}, V_{1:N,1:t})$$
$$- \mu_{\theta,t}^2$$

(125)

While fitting a Gaussian to the sum of two Gaussians could be a very in-exact approximation, in

practice the two individual distributions are close enough for this not to be a problem (as any contribution from $A_t$ to the posterior of $\theta_t$ is very small).

In conclusion, subjects report $\hat{S}_t$ (through a button response, see *Equation 121*) and they propagate the posterior $P(\theta_t|A_{1:t}, V_{1:N,1:t})$ (see *Equation 123*) as prior for the next trial.

