## [Decision Letter]

**Acceptance summary:**

Our perception is notoriously inaccurate, and estimating the uncertainty of a percept is an important task of our brain. The present paper shows for the first time that our brain uses past history in the estimation of the uncertainty of a current percept. It opens up further research questions including those related to the role of learning in perception.

**Decision letter after peer review:**

Thank you for submitting your article "Using the past to estimate sensory uncertainty" for consideration by *eLife*. Your article has been reviewed by three peer reviewers, including Tobias Reichenbach as the Reviewing Editor and Reviewer #1, and the evaluation has been overseen by Andrew King as the Senior Editor. The following individual involved in review of your submission has agreed to reveal their identity: Luigi Acerbi (Reviewer #3).

The reviewers have discussed the reviews with one another and the Reviewing Editor has drafted this decision to help you prepare a revised submission.

Summary:

Beierholm et al. present a well-executed psychophysical study in which participants judged the location of a conflicting visual-auditory stimulus under varying degrees of visual noise. Unlike other studies in this field, the visual noise was varied slowly, enabling participants to take advantage of this fact by incorporating estimates of uncertainty from the recent past into their current cue-weighting strategy. The authors then compare the data to three computational models: a simple learner that does not use past information, a Bayesian learner, and an exponential learner that accounts for past information at a fixed learning rate. The conclusion of the paper is that subjects' estimate of visual variability is influenced by the past history of visual noise.

While we find these results intriguing, and the experiment and analysis thorough, we have several major comments that we would like the authors to address.

Essential revisions:

1) The current analysis does not consider the effect that past visual locations, in addition to the uncertainty in the visual signal, may have on the estimate of the current location. In particular, if the location at t-1 is the same or close to the location at time t, it might be that past estimates of location get integrated (in fact, this becomes another causal inference problem). This effect might be further related to the question investigated here, because, if present, it might be modulated by the visual noise in the previous trial. Have you investigated whether this effect is present, and if so, how did you account for it in your analysis?

2) The authors currently refer to a preprint of this manuscript on bioarxiv for a full derivation of both model components for the Bayesian learner. Instead, please provide the full model derivation in the supplementary information in a self-contained form. Please also make the code for the modelling and the analysis as well as the data publicly available.

3) In the sinusoidal condition the bins had a duration of only 1.5 s, but the trials were 1.4 to 2.8 s apart. It therefore appears as if there were either 1 or 0 (or very rarely 2) responses in each bin. How did you handle the zero-response bins? And how can weights – presumed to vary smoothly between 0 and 1 – be reliably estimated from a single behavioural response?

4) The computational model makes certain assumptions that appear to differ from the experiment. We would like the authors to comment on these discrepancies. First, the computational model assumes that the auditory signal follows a normal distribution around a particular mean – . However, in the experiment, the location of the sound was either +5 degrees or -5 degrees away from the mean of the visual signal. Second, regarding the computational model, the authors write that "the dispersion of the individual dots is assumed to be identical to the uncertainty about the visual mean, allowing subjects to use the dispersions as an estimate of the uncertainty about the visual mean". But in the experiment there is no notion of an uncertainty (noise) in the visual mean. Third, the authors write that all probabilities, except for one, are Gaussian. As for the first point raised above, in the experiment, this only seems true for the distribution of the dots around the mean, but not for the other distributions. In particular, the mean of the visual signal is sampled from a discrete uniform distribution that encompasses only five different locations. Fourth, each dot location V_i,t_ is drawn from a normal distribution with mean U_t_, but U_t_ is drawn from another distribution with mean S_V,t_ – are the variance of these two distributions the same? Wouldn't U_t_ simply be the location (-10, -5, etc) on that trial, and wouldn't this mean instead that the dot positions are doubly stochastic? If so, why? The actual dispersion (not to mention the observers' estimates thereof) would be very noisy if dot locations were simply resampled at 5 Hz from a fixed distribution for a given trial. Doesn't resampling the SD at 5 Hz just complicate the modeling even more than it already is? Please also explain the purpose of the log random walk.

[Editors' note: further revisions were suggested prior to acceptance, as described below.]

Thank you for resubmitting your article "Using the past to estimate sensory uncertainty" for consideration by *eLife*. Your revised article has been reviewed by three peer reviewers, including Tobias Reichenbach as the Reviewing Editor and Reviewer #1, and the evaluation has been overseen by Andrew King as the Senior Editor. The following individual involved in review of your submission has agreed to reveal their identity: Luigi Acerbi (Reviewer #3).

The reviewers have discussed the reviews with one another and the Reviewing Editor has drafted this decision to help you prepare a revised submission.

Summary:

The authors have addressed our previous comments well in their extensively revised version of the manuscript. We only have a few remaining queries.

Revisions:

Difference in STD between current and previous bin predicts auditory weight, would that be expected given the autocorrelation of the STD sequence? Our intuition is that the null hypothesis (no impact of previous bin) would only be valid if STDs were (temporally) conditionally independent. In other words, if it were only the current STD that affected the weight, you might still see an apparent influence of the previous STD simply because STD of bin N is highly correlated with STD of bin N-1 (for the sinusoids at least).

This logic may only apply if the regression was on the absolute STDs, not the difference between current and previous STD, which is what the authors did. So perhaps it's not an issue. But if it is, we think one could perform a nested model comparison to test whether adding the previous time bin significantly improves the fit enough to justify the extra parameter. (It could also be that the current analysis is effectively doing this.)

Alternatively, one could perform this analysis separately for the first half vs. second half and see whether you observe a change in the regression coefficient for the δ-STD. If the authors' interpretation is correct, the coefficient should systematically change (sign flip?) when STD is increasing vs decreasing, whereas if the autocorrelation were driving its significance, it should not depend on increasing vs decreasing.

---

## [Author Response]

Essential revisions:1) The current analysis does not consider the effect that past visual locations, in addition to the uncertainty in the visual signal, may have on the estimate of the current location. In particular, if the location at t-1 is the same or close to the location at time t, it might be that past estimates of location get integrated (in fact, this becomes another causal inference problem). This effect might be further related to the question investigated here, because, if present, it might be modulated by the visual noise in the previous trial. Have you investigated whether this effect is present, and if so, how did you account for it in your analysis?

We thank the reviewer for this suggestion. To quantify the influence of the previous visual location, we expanded our regression model by another regressor modelling the visual cloud’s location on the previous trial. For instance, for bin = 1 we computed:

R_A,trial, bin=1_= L_A,trial,bin=1_* ß_A,bin=1_ + L_V,trial,bin=1_* ß_V,bin=1_ + L_V,trial-1,bin=1_* ß_Vprevious,bin=1_ + ß_const,bin=1_ + e_trial,bin=1_ with R_A,trial, bin=1_ = Localization response for current trial that is assigned to bin 1; L_A,trial,bin=1_ or L_V,trial,bin=1_= ‘true’ auditory or visual location for current trial that is assigned to bin 1; L_V,trial-1,bin=1_ ‘true’ visual location for corresponding previous trial (for explanatory purposes, we assign the bin of the current trial; the previous trial actually falls into a different bin); ß_A,bin=1_ or ß_V,bin=1_ = auditory or visual weight for bin = 1; ß_Vprevious,bin=1_ quantified the influence of the visual location of the previous trial on the perceived sound location of the current trial for bin 1. ß_const,bin=1_ = constant term; e_trial,bin=1_ = error term.

This analysis indeed reveals that the location of the visual cloud on the previous trial influences observers’ perceived sound location (Supplementary file 1—table 2). But surprisingly, it has a repellent effect, i.e. observers’ perceived that sound location shifts away from the true visual location. Importantly, having regressed out the influence of the previous V location on observers’ perceived sound location, we have repeated our main analyses, i.e. the repeated-measures ANOVA assessing whether w_A,bin_ differed for the bins in first vs. second half (see Supplementary file 1—table 3). Moreover, we repeated the regression model analysis to assess whether w_A,bin_ was predicted not only by the cloud’s STD of the current, but also by the previous bin (Supplementary file 1—table 4). Both analyses replicated our initial findings.

In addition, we also demonstrated that the regression weight quantifying the influence of the previous visual location did not correlate with the visual noise in the current trial r(ß_Vprevious,bin_, σ_Vcurrent,bin_) and in the previous trial (r(ß_Vprevious,bin_, σ_Vprevious,bin_)) (see Supplementary file 1—table 2).

We have now included additional methods in Appendix 1, report those results in Supplementary file 1—table 2-4 and Figure 2—figure supplement 1 and refer to the control analyses in the main text.

2) The authors currently refer to a preprint of this manuscript on bioarxiv for a full derivation of both model components for the Bayesian learner. Instead, please provide the full model derivation in the supplementary information in a self-contained form. Please also make the code for the modelling and the analysis as well as the data publicly available.

We have now added the full model derivation to the Appendix 2. Further, we uploaded the code for modelling and analyses scripts along with the behavioral data and model predictions to an OSF repository: https://osf.io/gt4jb/

We refer to this website in the main text.

3) In the sinusoidal condition the bins had a duration of only 1.5 s, but the trials were 1.4 to 2.8 s apart. It therefore appears as if there were either 1 or 0 (or very rarely 2) responses in each bin. How did you handle the zero-response bins? And how can weights – presumed to vary smoothly between 0 and 1 – be reliably estimated from a single behavioural response?

We are sorry for this confusion. Indeed, the reviewer is absolutely right. The bins in the four sequences had durations (Sin, RW1 = 1.5s, RW2 = 6s, Sin Jump = 2s) which were partially shorter than the ITI of 1.4-2.8 s, so that during the presentation of a single sequence bins had only 0,1 or rarely 2 responses. However, the experiment looped multiple times (Sin, RW1, Sin Jump ~ 130x, RW2 ~ 32) through the sequences during the course of the experiment. As a result of the jittered trial onset asynchrony, trials sampled different bins over replications/cycles of the same sequence throughout the experiment. In fact, each time bin was informed by at least 44-87 trials (see Supplementary file 1—table 1). Thus, the auditory weights could be estimated quite reliably.

We have now described the experimental design and analysis strategy in greater detail in the Results and Materials and method sections. We have also introduced a new notation for the equations and parameters for clarification. Moreover, we have included the following table into Supplementary file 1

4) The computational model makes certain assumptions that appear to differ from the experiment. We would like the authors to comment on these discrepancies.

Thank you for giving us the opportunity to motivate the assumptions of our model. We have now clarified the models’ assumptions in the Materials and method section.

First, the computational model assumes that the auditory signal follows a normal distribution around a particular mean. However, in the experiment, the location of the sound was either +5 degrees or -5 degrees away from the mean of the visual signal.

Indeed, the reviewer is absolutely right that the sounds are ± 5° from the visual location. However, observers are known to be limited in their sound localization ability, particularly if sounds do not come from natural sound sources but are generated with generic head-related transfer functions as in our study. Given observers’ substantial spatial uncertainty when locating sounds, we feel the model’s normal assumptions about sound location can be justified.

Second, regarding the computational model, the authors write that "the dispersion of the individual dots is assumed to be identical to the uncertainty about the visual mean, allowing subjects to use the dispersions as an estimate of the uncertainty about the visual mean". But in the experiment there is no notion of an uncertainty (noise) in the visual mean.

We have introduced the additional hidden variable U_t_ to account for the fact that even when auditory and visual signals come from a common source, they do not necessarily fully coincide in space in our natural environment. This introduces additional spatial uncertainty, so that observers cannot fully rely on the visual cloud of dots to locate the sound even in the common source situation. Critically, – as cited by the reviewer- because the dispersion of the dots and the uncertainty about the mean were set to be equal, observers could estimate this visual uncertainty from the spread of the dots.

Third, the authors write that all probabilities, except for one, are Gaussian. As for the first point raised above, in the experiment, this only seems true for the distribution of the dots around the mean, but not for the other distributions. In particular, the mean of the visual signal is sampled from a discrete uniform distribution that encompasses only five different locations.

Again, given the uncertainty about visual location this seems like a justifiable assumption. In fact, this assumption has been made by a growing number of studies that fitted the Bayesian Causal Inference model to observers’ localization responses, even though in all of those previous studies (Kording et al., 2007, Rohe and Noppeney, 2015b, Rohe and Noppeney, 2015a), the mean of the visual and auditory signals were sampled from a discrete uniform distribution.

Fourth, each dot location V_i,t_ is drawn from a normal distribution with mean U_t_, but U_t_ is drawn from another distribution with mean S_V,t_ – are the variance of these two distributions the same?

Yes – as explained in response to point 2, otherwise they would not be informative

Wouldn't U_t_ simply be the location (-10, -5, etc) on that trial, and wouldn't this mean instead that the dot positions are doubly stochastic? If so, why?

No, U_t_ is not identical to the location on that trial but is modelled as sampled from a Gaussian centred on this location; so in this sense, the model make the inference doubly stochastic; but importantly the two standard deviations are the same, so one is informative about the other one.

The actual dispersion (not to mention the observers' estimates thereof) would be very noisy if dot locations were simply resampled at 5 Hz from a fixed distribution for a given trial. Doesn't resampling the SD at 5 Hz just complicate the modeling even more than it already is?

It is true that the SD was resampled at 5Hz in order to provide the observer with the impression of a continuous stimulus. But for modelling, we focused selectively on the SD of the visual cloud at the trial onset times.

Please also explain the purpose of the log random walk.

We performed a random walk on the logarithm of the visual reliability λV,t as a convenience for the modeling. Previous research in the reward learning domain has compared a log random walk with a change point model and found very similar results (Behrens et al., 2007). Moreover, as even the exponential learner provided a reasonable fit to the data, we suspect that this type of assumption is unlikely to have a relevant effect.

As mentioned above, we included a critical discussion of our modelling assumptions in the Materials and methods section in which we discuss the aforementioned points.

[Editors' note: further revisions were suggested prior to acceptance, as described below.]

Revisions:Difference in STD between current and previous bin predicts auditory weight, would that be expected given the autocorrelation of the STD sequence? Our intuition is that the null hypothesis (no impact of previous bin) would only be valid if STDs were (temporally) conditionally independent. In other words, if it were only the current STD that affected the weight, you might still see an apparent influence of the previous STD simply because STD of bin N is highly correlated with STD of bin N-1 (for the sinusoids at least).This logic may only apply if the regression was on the absolute STDs, not the difference between current and previous STD, which is what the authors did. So perhaps it's not an issue. But if it is, we think one could perform a nested model comparison to test whether adding the previous time bin significantly improves the fit enough to justify the extra parameter. (It could also be that the current analysis is effectively doing this.)Alternatively, one could perform this analysis separately for the first half vs. second half and see whether you observe a change in the regression coefficient for the δ-STD. If the authors' interpretation is correct, the coefficient should systematically change (sign flip?) when STD is increasing vs decreasing, whereas if the autocorrelation were driving its significance, it should not depend on increasing vs decreasing.

Thanks for raising this point. Indeed, as the reviewer notes, the difference in STD between the current and previous bin is only weakly correlated to current STD ( r ~ 0.3 across the sequences). Further, because we inserted the difference in STD and the STD in the same regression model, each parameter estimate reflects only the unique variance that cannot be explained by any other regressor in the model. Hence, testing for the significance of one parameter estimate is equivalent to comparing two nested models that do or do not include this regressor. However, in the two-stage summary-statistic approach this relationship is less transparent.

Following the reviewer’s suggestions, we have now implemented a nested model comparison. We fitted two linear mixed effects models to the relative auditory weights: a full model with current STD and the difference in STD versus a reduced model with only current STD. The model comparison shows a greater model evidence for the full model.

We mention the control analysis in the Result section, added a supplementary table (Supplementary file—table 5) and describe the analysis in more detail in Appendix 1.